# EvidenceBench: A Benchmark for Extracting Evidence from Biomedical Papers

## Abstract

We study the task of automatically finding evidence relevant to hypotheses in biomedical papers. Finding relevant evidence is an important stage when humans write systematic reviews about certain scientific hypotheses. We introduce EvidenceBench to measure models performance on this task, which is created by a novel pipeline that consists of hypothesis generation and sentence-by-sentence annotation of biomedical papers for relevant evidence, completely guided by and faithfully following existing human experts judgment. Our pipeline's value and accuracy is validated by teams of human experts. We evaluate a diverse set of language models and retrieval systems on the benchmark and find the performance of the best models still falls significantly short of expert-level on this task. To show the scalability of our proposed pipeline, we create a larger EvidenceBench-100k with 107,461 fully annotated papers with hypotheses to faciliate model training and development. Both datasets are available at https://github.com/EvidenceBench/EvidenceBench.

## 1 Introduction

Systematic reviews serve as a cornerstone of evidence-based research across scientific disciplines, offering comprehensive syntheses of current research directions on specific questions or topics. Their impact is particularly pronounced in biomedicine, where they play a critical role in evaluating the efficacy of therapeutic interventions and informing clinical practice. These reviews play a crucial role in shaping healthcare policies (Bunn et al., 2015) adopted by national agencies in countries like the US (Viswanathan et al., 2012), UK (Alderson & Tan, 2011), and Australia (NHMRC, 2019). They are widely utilized by universities, hospitals and research institutions. Each review requires intensive labor from human experts and costs approximately 141 thousand US dollars to produce. Annually, major pharmaceutical companies and universities each spend over 18 million US dollars to produce these systematic reviews (Michelson & Reuter, 2019). Leading systematic review organizations, such as The Cochrane Collaboration (2023), have established sustainable business models, generating annual revenues exceeding 20 million US dollars from licensing and royalties, underscoring both the economic value and essential nature of high-quality evidence synthesis in biomedicine.

With over 1 million biomedical research papers published each year and more than 35 million papers already archived in the PubMed database of biomedical literature (González-Márquez et al., 2024), it has become increasingly labor-intensive for humans to create systematic reviews for emerging questions and hypotheses. Consequently, there is significant research and commercial interest in developing automated systems that can assist—and potentially one day replace—humans in creating systematic reviews.

Biomedical systematic reviews follow a three-stage methodology (Higgins & Green, 2011): *search*, *extract*, and *analyze*. The *search* stage leverages structured medical databases like MEDLINE (2024) through keyword queries to identify candidate papers. The *analyze* stage employs specialized statistical software such as RevMan (Higgins & Green) for quantitative synthesis and visualization of findings. While these stages present opportunities for algorithmic advancement, we focus on automating the crucial *extract* stage—the identification of hypothesis-relevant evidence from biomedical literature. The extract stage, despite being fundamental to the review's quality and currently requiring intensive manual effort, remains largely unaddressed by computational approaches.

There is currently no established benchmark or training resource for this task. To address this gap, we introduce EvidenceBench (see Figure 1), a benchmark designed to evaluate models' abilities

to extract evidence relevant to a given hypothesis. Additionally, we present EvidenceBench-100k, which serves both as a benchmark for model performance and as a resource to aid in the training and development of models for this task. Our proposed datasets are fully open-sourced under the CC-BY license and encompass a comprehensive range of biomedical topics.

We introduce a novel pipeline for creating EvidenceBench and EvidenceBench-100k, powered by Large Language Models (LLMs). The pipeline has two main components: hypothesis generation and an alignment annotator that matches hypotheses with sentences from papers. Informally, we use existing **evidence summary** from review papers to generate a hypothesis and use the same **evidence summary** to find sentences that provide evidence for the generated hypothesis. See Section 3 and Figure 2 3 for details.

Our pipeline is highly scalable, reducing the construction time of EvidenceBench from over 3,000 human hours and $120,000 in wages to just 3 API hours and $5,000 in API costs, using state-of-the-art LLMs at the time of data creation, Claude3-Opus for hypothesis generation and GPT4-0125 for alignment annotation. During the construction of EvidenceBench-100k, we used GPT4-o-mini for alignment annotation and kept the construction time and cost under 24 hours and $5000.

Table 5b demonstrates EvidenceBench-100k is suitable for fine-tuning LLMs and embedding models as we observed significant improvements of fine-tuned models over their pretrained baselines.

We conduct a benchmarking study on a variety of state-of-the-art large language models (LLMs) and embedding models, which provide several insights. First, although the best LLMs still fall short of expert-level performance on this task, indicating that they cannot replace humans, they have the potential to assist them. Second, embedding models consistently underperform compared to large language models due to lack of global context from research papers. Third, we report that current LLMs trained for long document understanding still get "Lost in the Middle" (Liu et al., 2024).

We highlight some important sections in the paper:

- Section 2.1 explains two critical concepts: **Study Aspect** and **Source of Information**.
- Section 2.3 presents Aspect Recall, the primary evaluation metric used throughout our experiments.
- Figure 2 illustrates our key methodological contribution: leveraging expert-written Evidence Summaries from review papers to guide an LLM-based annotation process (Figure 3). This ensures that our pipeline adheres to human expert judgments.
- Section 3.3.1 presents expert panel validation confirming the quality and research merit of our generated hypotheses.
- Section 3.4 and Table 2 provide statistical evidence that our GPT4 annotation procedure achieves comparable performance to biomedical PhD students.

## 2 TASK FORMULATION

The EvidenceBench task is to identify the most important pieces of evidence relevant to a hypothesis. This is formulated as a sentence retrieval task. Given a paper, the task is to retrieve a set of sentences that jointly provide the most important pieces of evidence. See Figure 1 for an example.

### 2.1 DEFINITIONS

**Candidate Pool:** The full-text of a research paper is presented as a list of sentences (including section and subsection headings but excluding figure and table and their captions). This ordered list of sentences is the Candidate Pool. Very importantly, research papers and review papers are completely different.

**Evidence Summary:** An evidence summary is written by human experts. It is included in open-sourced review papers, such as surveys, monographs and systematic reviews. An evidence summary is directly linked to one single research paper. It contains all pieces of evidence (from this research paper) that the human experts believe are important and relevant to a hypothesis. Evidence summary could take the form of a normal paper summary, or a bulleted list, or even tabular format.

**Study Aspect:** A study aspect, denoted as $f_i$, is a single piece of information. For instance, an experimental outcome or a detail from study design. Note, study aspects are decomposed from a evidence summary written by experts. Each study aspect is identified by human experts and represent an important detail relevant for a hypothesis. Therefore, study aspects are considered to be human experts' judgment.

**Source of Information:** A sentence in a research paper is considered a source of information for a study aspect if it satisfies the following two criteria.

1. The content of the sentence implies most of the study aspect.

2. For any part of the study aspect that the sentence does not cover, the information must be easily deducible from the surrounding context.

Given a sentence $s_j$ and a study aspect $f_i$, define the source-of-information indicator function $\mathcal{S}(f_i, s_j)$:

$$\mathcal{S}(f_i, s_j) = \begin{cases} 1 & \text{if } s_j \text{ is a the source of information for } f_i \\ 0 & \text{otherwise} \end{cases}$$

Note, if a sentence is a source of information for a study aspect, we informally say the sentence covers this aspect. Note, since study aspect is decomposed and only represent one piece of information, one single sentence is enough to cover it.

**Hypothesis:** A hypothesis is a scientific generalization, usually expressible in one line (less than 50 words). It should not be tied to specific details of an experiment. See Figure 1 for an example and Section 3.3 for the generation and validation process.

**Evidence Set:** The evidence set is the set of study aspects which provides evidence relevant to a hypothesis. A study aspect in this set may provide evidence on its own, or only in combination with other study aspects. Very importantly, we use an **Evidence Summary** to derive an evidence set, see Figure 2 right side.

## 2.2 TASK DEFINITION

We now introduce our primary task, Evidence Retrieval @K (ER@K). Informally, the task is to find K sentences in a research paper which provide the greatest amount of evidence relevant to a hypothesis. This is operationalized as finding sentences in the research paper which cover the most study aspects from the evidence set.

Formally, given a hypothesis and a candidate pool, the task for a system is to retrieve K sentences from the candidate pool which provide evidence relevant to the hypothesis. The retrieved sentences are then evaluated against the evidence set, which contains ground-truth study aspects (i.e. pieces of evidence relevant to the hypothesis identified by human experts). The goal is for the K retrieved sentences to be sources of information for as many of the study aspects in the evidence set as possible. Crucially, during the retrieval task, the system does not have access to the ground-truth evidence set; the evidence set is only used for evaluation.

In the second version of the task, only study aspects related to the results and analyses are considered. Study aspects related to background and methods are filtered out of the evidence set. This task is called Result-ER@K, and focus on system's ability to identify numerical and experimental results.

## 2.3 EVALUATION METRICS

To determine the quality of a system's retrieved sentences, we use **Aspect Recall**. Let $\{s_1, \ldots, s_k\}$ be the set of retrieved sentences, and $\{f_1, \ldots, f_m\}$ be the set of study aspects in the evidence set. The Aspect Recall is defined as

$$\frac{\sum_{f_j} \mathbf{1}_{\left(\sum_{s_i} \mathcal{S}(s_i, f_j)\right) \geq 1}}{m}$$

This measures the fraction of study aspects that can be covered by a retrieved set of k sentences. See Figure 1 for an example calculation of Aspect Recall.

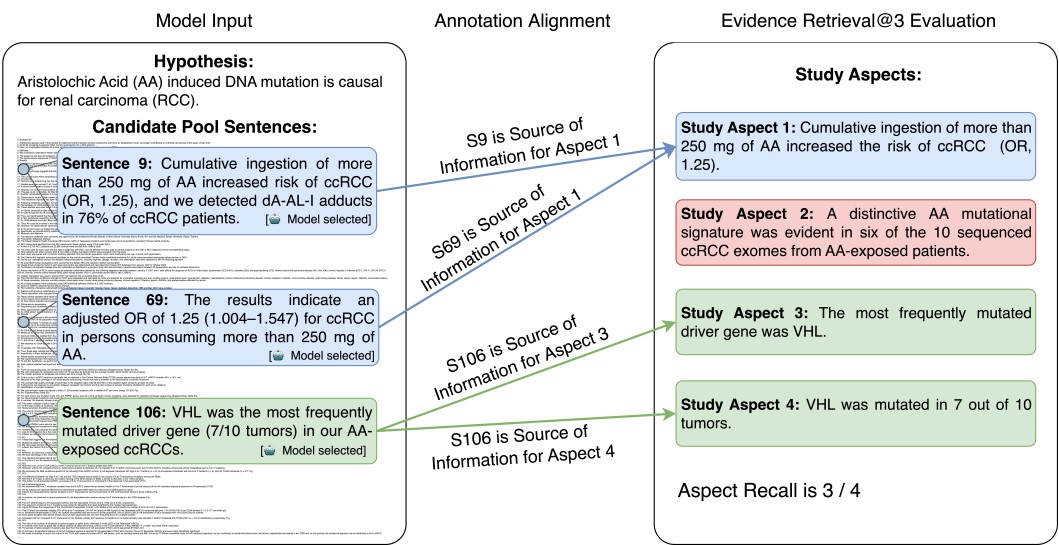

Figure 1: In EvidenceBench, a model sees a Hypothesis, and the full sequence of sentences from a paper as Candidate Pool. The model selects S9, S69, and S106 as the set of retrieved sentences. However, these 3 sentences only cover Aspect 1, 3, and 4, since S9 and S69 are redundant and cover the same Aspect 1. Aspect 2 is missed, resulting in 75% Aspect Recall.

## 3 DATASET CONSTRUCTION PIPELINE

### 3.1 DATA SOURCES

There are two data sources for EvidenceBench and EvidenceBench-100k. First, a collection of 107,887 CC-BY open-sourced biomedical research papers where each research paper represents a datapoint. Second, a collection of 44,772 review papers from PubMed Central. Each biomedical research paper has a corresponding evidence summary included in one review paper. Specifically, EvidenceBench has 426 datapoints and EvidenceBench-100k has 107,461 datapoints.

#### 3.1.1 TRAIN/TEST SPLIT

We use a train/test split of 133/ 293 task instances for the original EvidenceBench. We use a train/test split of 87,461/20,000 for EvidenceBench-100k. All of the prompt optimization is performed on the train sets and all of the few-shot examples used in the prompts are selected from the train sets. All datasets have CC-BY licenses.

Table 1: EvidenceBench Test Set. Optimal Number of Sentences refers to the smallest number of sentences that are sources of information for the most of study aspects in an evidence set.

| Dataset | n | Candidate Tokens | | | Sentences | | | Study Aspects | | | Optimal Number of Sentences | | |
|---|---|---|---|---|---|---|---|---|---|---|---|---|---|
| | | min | avg | max | min | avg | max | min | avg | max | min | avg | max |
| Test Set | 293 | 1691 | 5578 | 23980 | 48 | 168.1 | 794 | 2 | 9.5 | 36 | 1 | 4.6 | 18 |
| Test Set (Result Retrieval) | 288 | 1691 | 5582 | 23980 | 48 | 168.4 | 794 | 1 | 4.2 | 18 | 1 | 2.1 | 7 |

### 3.2 DATASET PIPELINE OVERVIEW

A task instance (i.e. a datapoint) is constructed as follows. Given an expert-written evidence summary, we generate a hypothesis from it. Since the evidence summary was written to summarize a research paper, we take the sentences from this research paper as the candidate pool for the task instance. We then decompose the expert-written evidence summary into a set of study aspects, also known as the evidence set. Each study aspect is used to annotate each sentence in the candidate pool. This is the alignment annotation process, which determines sentences in the research paper that are sources of information for the study aspect, and consequently, are relevant for the hypothesis. In the next

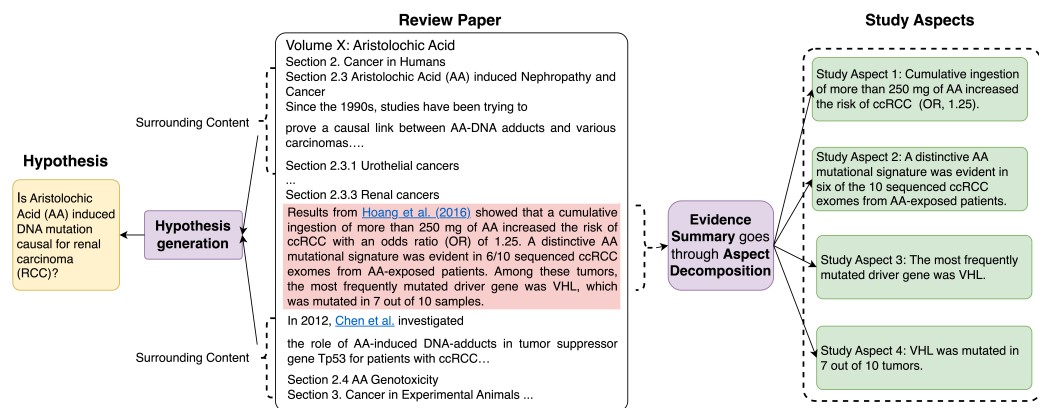

Figure 2: The highlighted paragraph in the review paper is the evidence summary. The left side shows how a hypothesis is generated (extracted) from the evidence summary and its surrounding context in the review paper. The right side shows the evidence summary being decomposed into study aspects.

sections, we explain the procedures of harvesting evidence summary from review papers, hypothesis generation, aspect decomposition, and alignment annotation.

**Harvesting Evidence Summary:** The highlighted portion in the review paper in Figure 2 is an evidence summary. Evidence summary has an XML citation embedded in it, so it can be identified and harvested by a simple deterministic algorithm. Further filtering and preprocessing is done by GPT4-0125 to make sure the extracted summary has high quality. A random sample of 50 extracted summaries are manually reviewed by researchers and confirmed the accuracy is 98%. See Appendix G for details.

**Aspect Decomposition:** For each evidence summary, we decompose it into study aspects. These study aspects comprise the evidence set for the research paper. On average, a summary can be broken down into 10 study aspects. In EvidenceBench, decomposition is done by GPT4-0125 and inspected by human researchers. In EvidenceBench-100k, the decomposition is done by GPT-4o-mini-0718 and 200 randomly sampled instances are inspected by human researchers and are found to be of high quality.

### 3.3 HYPOTHESIS GENERATION

A review paper focuses on a specific hypothesis and survey a number of research papers, summarizing the evidence that each provides for the hypothesis. For each evidence summary, our goal is to extract the hypothesis that it is providing relevant evidence to. Think of a hypothesis as an explicit or implicit question waiting to be recovered. In order to do this, we provide an LLM (Claude3-Opus) with the evidence summary as well as surrounding paragraphs. The model is then prompted to recover the hypothesis being discussed in the review paper. See Figure 2 left side. To ensure high quality hypotheses for EvidenceBench-100k, Claude3-Opus is also used.

#### 3.3.1 EXPERT VALIDATION OF HYPOTHESES

We perform an expert evaluation of the hypotheses extracted from review papers in EvidenceBench, focusing on two questions:

1. Does the hypothesis have sufficient scientific value?

2. Does the corresponding evidence summary provide evidence which is relevant to the hypothesis?

The annotation team for this task consisted of three medical doctors. The first expert defined the annotation guidelines and provided feedback on an initial set of 20 extracted hypotheses. This feedback was also used to perform prompt optimization for Claude3-Opus.

After finalizing the guidelines and prompt, a separate set of 50 hypotheses was generated. The two other annotators each evaluated 25 hypotheses. Annotation guidelines are in Appendix B.

For Question 1, 50/50 hypotheses were judged to have sufficient scientific value. For Question 2, 47/50 hypotheses were judged to be relevant to the corresponding evidence summaries. This demonstrates that hypotheses were correctly extracted from review papers.

### 3.4 ALIGNMENT ANNOTATION OF STUDY ASPECTS AND SENTENCES

We have so far described the procedure for decomposing study aspects and recovering hypotheses from the review papers. A list of study aspects describes the evidence that a specific research paper provides relevant to a hypothesis. The final step is to identify which sentences of the original research paper serve as sources of information for each study aspect. Because study aspects can, in general, come from any part of the research paper, this requires annotating every sentence in the research paper according to whether it matches each study aspect. See Figure 4 for the distribution of relative positions of sentences that are sources of information for some aspects.

Research papers in the dataset have approximately 168 sentences and 10 study aspects on average. EvidenceBench contains more than 400 research papers. Sentence-by-sentence annotation requires approximately 700,000 sentence annotations, which is infeasible given the use of expert annotators; we estimate that it would require more than 3000 hours of annotation. We therefore develop a pipeline for automating the annotation process, and perform human evaluation of its reliability. We observe that the task of labeling a sentence according to whether it is a source of information for a study aspect is considerably simpler than the benchmark's full sentence retrieval task. It only requires a judgment of whether a single sentence from the research paper contains most of the same information as a study aspect.

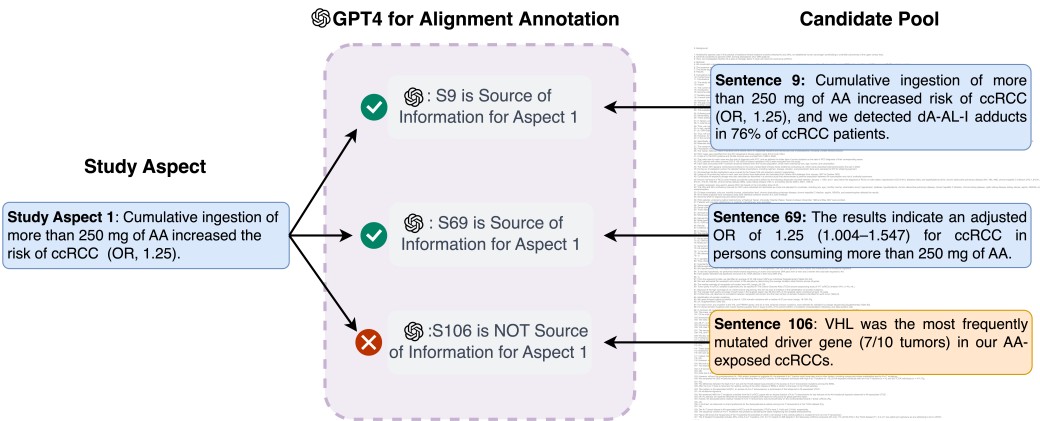

Figure 3: The process for matching sentences with study aspects. In this example, GPT4 sees the study aspect, one candidate sentence from the research paper with the context, and is prompted to determine whether the candidate is a source of information for the study aspect.

For the annotation pipeline, GPT-4 is shown a target sentence from the research paper and a study aspect (as well as some additional context: the 10 surrounding sentences from the research paper, and the evidence summary from the review paper). It is then asked to evaluate whether the target sentence implies most of the information contained in the study aspect. See Figure 3.

The optimization and evaluation of the pipeline were performed using a development set/test set split. The prompt and annotation methodology were optimized on a development set of 37 research papers. The prompt and optimization procedure is provided in Appendix C.

After the pipeline was finalized, it was evaluated on a test set of 50 randomly sampled research papers. For each research paper, a single study aspect was selected, and every sentence in the paper was annotated for this study aspect. The labeling was performed by four annotators, who are Ph.D. researchers in bioinformatics. The annotators were split into two teams. Each annotator first performed the annotation task independently. The pairs within each team then consulted with each

other to reach consensus judgments. Finally, inter-annotator agreement was calculated by comparing the judgments of the two teams. Full annotation guidelines are provided in Appendix B.

Each team labeled 8111 (sentence, study aspect) pairs in total. Table 2 shows inter-annotator agreement between the two human teams, and between the human teams and GPT-4. Human and GPT-4 judgments match each other more than 98% of the time. Because of class imbalance (positive labels are rare, around 150 out of 8111), other measures of agreement such as Cohen's $\kappa$ are in the mid 60's, indicating substantial agreement between the human teams and between the human teams and GPT-4. Bootstrapped hypothesis tests find no significant difference between the human/human agreement rate and the human/GPT-4 agreement rate.

For the larger EvidenceBench-100k, over 150 million sentence-aspect pair judgements would need to be made. We switch the annotator to GPT4-o-mini-0718. We validated the quality of its annotation by re-running the hypothesis test on the same 50 randomly sampled papers, and found very similar Human & GPT agreement, passing all hypothesis tests.

Note, the novel annotation alignment procedure and the statistical hypothesis testing are our major contributions to the research practice of synthetic data generation.

Table 2: Hypothesis Testing Results for Automatic Alignment Annotation

| Metrics | Human & Human Average | Human & GPT Average | p-value |
|---|---|---|---|
| Exact Accuracy | $98.8 \pm 0.3$ | $98.7 \pm 0.2$ | 0.21 ✓ |
| F1 Binary | $66.0 \pm 6.5$ | $64.6 \pm 5.6$ | 0.64 ✓ |
| Cohen's $\kappa$ | $65.4 \pm 6.6$ | $63.9 \pm 5.6$ | 0.63 ✓ |
| Spearman's $\rho$ | $65.4 \pm 6.6$ | $64.0 \pm 5.6$ | 0.65 ✓ |

## 4 EXPERIMENT SETUP

**EvidenceBench Tasks:** we consider four evidence retrieval tasks with different settings.

**Evidence Retrieval @Optimal:** denoted as ER@optimal. The smallest number of sentences to cover all aspects is denoted as Optimal. The average optimal number is only 4.6. The task is for a model to retrieve no more than the Optimal number of representative sentences to form the best evidence set relevant to the hypothesis.

**Evidence Retrieval @10:** denoted as ER@10. The task is to retrieve no more than 10 sentences relevant to the hypothesis. Since there are rare instances where the optimal number is more than 10, the maximum possible performance for this task is 99.3%.

**Result Evidence Retrieval @Optimal:** denoted as Result-ER@Optimal. The task restricts the model to retrieve no more than the optimal number of sentences, which is calculated by the minimum number of sentences required to cover all study aspects labeled as "Results". This labeling is done by GPT4-0125. Empirically, half of the aspects are labeled as "Results', see Table 1.

**Result Evidence Retrieval @5:** denoted as Result-ER@5.

**Experiment Models** We test the following models Claude3-Opus (Anthropic, 2024), Gemini 1.5 (Google, 2024), GPT-4o (OpenAI, 2024), Llama3-70B, Llama3-8B (AI, 2024), E5-v2 (Wang et al., 2022), OpenAI Embedding v3 (OpenAI, 2024), VoyageAI v2 (Voyage AI, 2024), GritLM-7B (Muennighoff et al., 2024), E5-Mistral-7B (Wang et al., 2023), NV-Embed-v2(Lee et al., 2024) which is the leader of the Massive Text Embedding Benchmark MTEB (Muennighoff et al., 2022).

### 4.1 EVALUATION STRATEGIES

We implement standard practices for evaluating LLMs on long-context benchmarks (Bai et al., 2023; Zhang et al., 2024),

**Chain-of-Thought (CoT)** is our default evaluation strategy, optimized from our train set.

**In-Context-Learning (ICL):** from the train set, we randomly sample 8 example hypotheses and their corresponding ground-truth set of retrieved sentences, as per standard ICL practices (Wei et al., 2022; Min et al., 2022), in addition to our default CoT prompt.

**Section-by-Section:** we divide a research paper by its natural sections. In the first stage, LLM only retrieves from each section one at a time. In the second stage, all retrieved sentences are presented to the LLM and final selections are made.

**Evaluation Metric is Aspect Recall**. See Section 2.3. Note for Result-ER@K, $m$ is the number of aspects labeled as "Results". For other cases, $m$ is the total number of aspects.

Table 3: Four tasks are reported on EvidenceBench test set. For each model, the highest number is reported if multiple strategies are used.

|  | GPT-4o | Claude3 | Gemini | LLama3-70B | OpenAI | Voyage | GritLM | E5-Mistral | NV-Embed |
|---|---|---|---|---|---|---|---|---|---|
| ER@Optimal | **51.4** | 47.6 | 48.3 | 46.7 | 25.1 | 22.7 | 27.0 | 22.7 | 25.2 |
| ER@10 | **71.6** | 66.4 | 65.4 | 65.4 | 42.2 | 42.0 | 46.4 | 41.9 | 44.7 |
| Result-ER@Opt | **52.6** | 51.7 | 46.7 | 46.2 | 19.1 | 18.3 | 18.9 | 19.3 | 20.1 |
| Result-ER@5 | **70.8** | 68.7 | 65.4 | 63.7 | 33.1 | 31.9 | 39.1 | 33.6 | 35.6 |

## 5 RESULTS

From Table 3, we list the best performance for each model (LLM or embedding model) on the four constrained Evidence Retrieval tasks in the original EvidenceBench. GPT-4o consistently outperforms others across all tasks, while Gemini, Claude3-Opus, and Llama3-70B closely trail behind. Llama3-70B can only be evaluated using the Sec-by-Sec strategy due to its limited context window, but it shows robust performance across all tasks. There is a qualitative difference between LLMs and embedding models, partially because embedding models are not context-aware when calculating sentence embeddings. This invites future work on general-purposed context-aware embedding models.

**In-context learning:** From Table 4, we see an 8-shot ICL does not significantly alter performances for LLMs. In particular, GPT-4o and Gemini-1.5 slightly improve, while Claude3-opus slightly degrades. This indicates the primary difficulty is context-length and not a failure to understand task requirement, suggesting that ICL is less effective on long-context benchmarks.

**Section-by-Section Processing:** On the other hand, from Table 4, Sec-by-Sec considerably improves Gemini and Claude's performances, suggesting that the default longer-context version of the task hinders their retrieval abilities. Section-by-section is by far the most robust strategy observed here.

Table 4: Comparison of different strategies for LLMs on the EvidenceBench test set.

| Model | Baseline | | ICL | | Sec-by-Sec | |
|---|---|---|---|---|---|---|
|  | ER @ Optimal | ER@ 10 | ER @ Optimal | ER@ 10 | ER @ Optimal | ER@ 10 |
| GPT-4o | **48.1** | **69.6** | **51.4** | **68.7** | **50.9** | **71.6** |
| Claude3 | 41.1 | 53.6 | 38.3 | 55.4 | 47.6 | 66.4 |
| Gemini | 42.7 | 63.0 | 43.2 | 62.4 | 48.3 | 65.4 |
| Llama3-70B | - | - | - | - | 46.7 | 65.4 |

### 5.1 FINE-TUNING AND EVALUATION ON EVIDENCEBENCH-100K

EvidenceBench-100k is split into a 80k train set and a 20k test set. For cost reasons, we randomly sample 3000 datapoints from the 20k test set. Table 5a shows EvidenceBench-100k test set can be used to evaluate and clearly differentiate various models' performance.

Furthermore, we fine-tune two models: E5-v2 335M and Llama3-8B (sec-by-sec strategy) using the 80k training datapoints. We test them on the original EvidenceBench test set for the task of Result-ER@Optimal. We notice both fine-tuned models show significant improvements over their baselines as shown in Table 5b. See full details of fine-tuning at Appendix H.

Table 5: Comparison of Model Performances on EvidenceBench Datasets

(a) EvidenceBench-100k test set

| Model | Result-ER@Optimal |
| --- | --- |
| GPT-4o | **42.84**% |
| Claude3 | 35.12% |
| GritLM | 14.59% |
| OpenAI | 10.95% |

(b) EvidenceBench test set.

| Model | Result-ER@Optimal |
| --- | --- |
| Pretrained Llama3-8B | 35.8% |
| Finetuned Llama3-8B | **41.0**% |
| Pretrained E5-v2 | 15.2% |
| Finetuned E5-v2 | **32.9**% |

## 6 ANALYSES

**Effectiveness of Current LLM Solutions:** Table 3 clearly indicates that current embedding models are inadequate for assisting or replacing human experts in identifying relevant evidence for biomedical hypotheses. We now examine the best-performing LLM solution, GPT-4o. In the task ER@Optimal, GPT-4o retrieves an average of 4-5 sentences per hypothesis, according to Table 1. In this setting, GPT-4o achieves a 50% Aspect Recall, covering half of the study aspects identified by human experts. This demonstrates that current LLMs cannot fully replace human experts in finding relevant evidence for hypotheses.

Conversely, per the task definition for ER@10, models retrieve 10 sentences. In this setting, GPT-4o achieves an aspect recall of 70%. According to Table 1, a typical research paper contains 168 sentences. Therefore, instead of reviewing the entire research paper, humans can use the 10 sentences retrieved by GPT-4o as an efficient starting point, while searching for additional sentences that cover potentially missing aspects. This demonstrates that, in this setting, GPT-4o can meaningfully assist human experts in locating and presenting evidence from research papers for hypotheses, a crucial step in writing review papers.

**Embedding Models Underperform Generative Models:** From Table 3, we observe that the performance of embedding models falls significantly short of the performance of similar-sized generative models. The embedding models GritLM-7B, E5-Mistral-7B and the state-of-the-art NV-Embed-7B cover at most 20.1% of aspects, while the pretrained LLama3-8B covers 35.8% of aspects. The shortcomings of GritLM-7B and NV-Embed-7B suggest that a naive local embedding of sentences, without contextual awareness, is insufficient for this task. Our empirical observations confirm that reasoning beyond individual sentences is necessary to solve this task effectively. For instance, in Figure 1, Sentence 9 and Sentence 69 convey the same information and both address study aspect 1. Only by comparing them together (i.e., reasoning beyond a single sentence) can models eliminate one of these sentences to reduce redundancy.

**Section-level Reasoning is Sufficient:** Table 4 shows that retrieving evidence section-by-section (Sec-by-Sec) achieves strong performance on the task. With Sec-by-Sec, a model can only read one section at a time, preventing it from reasoning across multiple sections. The strong performance of this method indicates that global reasoning across the entire research paper is not essential for retrieving evidence. This can be explained by the structure and organization of biomedical research papers, where the content of each section is relatively self-contained, and interaction across sections is sparse. This suggests that LLMs trained for much longer contexts (e.g., over 10,000 tokens) may not be necessary for this task.

**LLMs Still Get "Lost in the Middle".** We observed that some papers have most of their important sentences concentrated at the beginning and end of the document, as shown in Figure 4. We categorize these papers into two groups. In the first category, the first 10 and last 10 sentences of a paper cover over 80% of the study's aspects. In the second category, these sentences cover less than 20% of the aspects. Out of 3,000 randomly sampled points from the EvidenceBench-100k test set, 1,115 papers

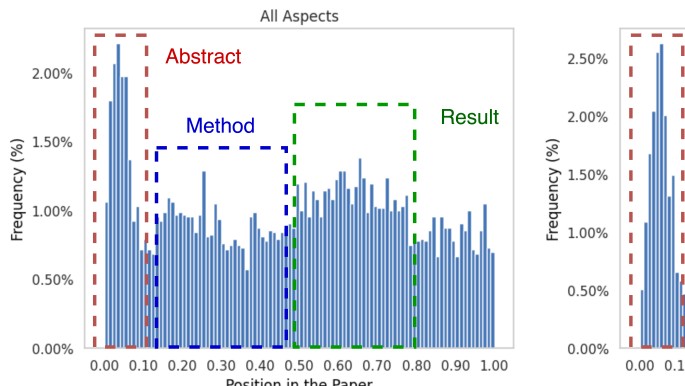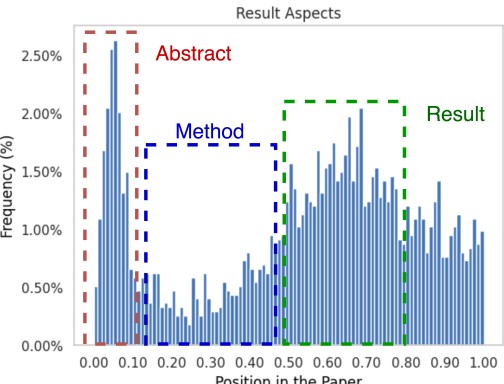

Figure 4: Original EvidenceBench. **Left** figure shows the distribution for the relative position in the candidate pool of all sentences that are considered as source of information for at least one study aspect. **Right** figure shows the same but for aspects labeled as 'Results'. Abstract sentences have much higher chance to be matched with aspects. However, all abstract sentences (around 10 sentences per research paper) only cover around 50% of all aspects, indicating no heuristic algorithm can cheat EvidenceBench.

fall into the first category, while 1,111 papers fall into the second category. GPT-4o's aspect recall performance is 51.6% in the first category and 34.9% in the second category. Claude3-Opus aspect recall is 42.8% in the first category and 28.2% in the second category. Note, during evaluation, we neither explicitly nor implicitly instruct LLMs to focus on any specific parts of the paper.

This indicates that LLMs perform significantly better when the important tokens are not located in the middle of the document. Our manual inspection also reveals that LLMs tend to focus on the beginning and end of the document. The "Lost in the Middle" phenomenon, as reported in previous LLMs (Liu et al., 2024), seems to persist in current LLMs on EvidenceBench.

## 7 RELATED WORK

**Hypothesis Generation** Recent works explore using LLMs to generate scientific hypotheses (O'Brien et al., 2024; Tong et al., 2023; Park et al., 2024; Baek et al., 2024; Abdel-Rehim et al., 2024). Qi et al. (2023) fine-tune LLMs on biomedical literature that pairs background knowledge with corresponding hypotheses, and then use the LLMs to generate hypotheses when prompted with background knowledge. In contrast, our hypothesis generation procedure is much more extractive in that it aims to extract existing implicit or explicit hypotheses from review papers, thus ensuring high alignment with domain experts' own perspectives.

**Evidence Retrieval** Claim-based retrieval (Chen et al., 2023) retrieve evidence by breaking down a complex claim into specific aspects and retrieving each aspect. On the other hand, our pipeline uses a novel approach, by decomposing summarized evidence from review papers into study aspects (instead of claims), which serves as ground-truth human domain experts knowledge that would guide our LLM-annotator to match sentences from research papers to these study aspects.

**LLM in Biomedicine** Researchers have shown strong performance of LLMs in BioNLP tasks, including relation extraction, question answering, document classification, name entity recognition, and summarization (Luo et al., 2022; Chen et al., 2024; Monajatipoor et al., 2024; Jahan et al., 2023; 2024; Munnangi et al., 2024). LLMs are also being used to extract specific information from report, e.g. Interventions, Outcomes, and Findings by Wadhwa et al. (2023).

## 8 CONCLUSION

We introduced EvidenceBench and EvidenceBench-100k, a benchmark for retrieving evidence for scientific hypotheses from biomedical literature. EvidenceBench was constructed using an automated, scalable pipeline that transforms expert-written summaries into fine-grained annotations linked to specific sentences in research papers.

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

**Appendix: Table of Contents**

# A    DATASET

## A.1    DATASET LICENSE AND CODE LICENSE

The EvidenceBench dataset uses the following licenses:

- Test set: Provided under CC-BY license.
- Train set: Provided under CC-BY-NC-SA license.
- Dev set: Provided under CC-BY-NC-SA license.
  The EvidenceBench-100 dataset uses CC-BY license.

A copy of the full license can be found at https://github.com/EvidenceBench/EvidenceBench/blob/main/LICENSE.md. Note that the test set has the most permissive license.

All code is released under the MIT License.    The full license can be found at https://github.com/EvidenceBench/EvidenceBench/blob/main/LICENSE.md.

## A.2    DATASET HOSTING, ACCESSIBILITY AND MAINTENANCE

The    EvidenceBench    and    EvidenceBench-100k    datasets    can    be    accessed    at (https://github.com/EvidenceBench/EvidenceBench).

## A.3    A MOTIVATING EXAMPLE

Aristolochic Acid (AA) is a toxin that is naturally occurring in traditional Chinese herbal medicines and has been known to cause many types of cancer in animals and humans Hoang et al. (2016). However, 20 years ago, the causal relationship between AA and kidney cancer was not yet confirmed. In this section, we present an example data instance related to AA.

**Hypothesis:**

Aristolochic Acid (AA) induced DNA mutation is causal for renal carcinoma (RCC).

**Evidence Summary**

Results from Hoang et al. (2016) showed that a cumulative ingestion of more than 250 mg of AA increased the risk of ccRCC with an odds ratio (OR) of 1.25. A distinctive AA mutational signature was evident in 6/10 sequenced ccRCC exomes from AA-exposed patients. Among these tumors, VHL, the most frequently mutated gene, mutated in 7 out of 10 samples.

**Study Aspect Decomposition:**

1. Cumulative ingestion of more than 250 mg of AA increased the risk of ccRCC (OR, 1.25). [Sentences 9 and 69 are sources of information for Aspect 1].

2. A distinctive AA mutational signature was evident in six of the 10 sequenced ccRCC exomes from AA-exposed patients. [Sentence 163 is the source of information for Aspect 2]

3. The most frequently mutated driver gene was VHL. [Sentence 106 is the source of information for Aspect 3].

4. VHL was mutated in 7 out of 10 tumors. [Sentence 106 is the source of information for Aspect 4]

**Full Paper:**

All 216 sentences from Hoang et al. (2016) are indexed, starting from 0 to 215. For brevity, we will not reproduce the entire paper here. Figure 1 shows sentences 9, 69, 106 from the full list of sentences are retrieved by a model.

**Selected Sentences from Full Paper:**

- Sentence 9: Cumulative ingestion of more than 250 mg of AA increased risk of ccRCC (OR, 1.25), and we detected dA-AL-I adducts in 76% of Taiwanese ccRCC patients.

- Sentence 69: The results (Table 1) indicate an adjusted OR of 1.25 (1.004–1.547) for ccRCC in persons consuming more than 250 mg of AA during the period of 1997 to 2003.
- Sentence 106: VHL was the most frequently mutated driver gene (7/10 tumors) in our AA-exposed ccRCCs (Table 2).
- Sentence 163: Whole-exome sequencing confirmed that the AA mutational signature was present in 6 of 10 ccRCC patients studied.

## A.4 DATASET COLLECTION AND PROCESSING

We use BioC API to download biomedical papers which are available in the PMC database. Papers unavailable in PMC are manually downloaded. We use GROBID to parse papers from PDF format to XML format. We use Stanza to split paragraphs of text into sentences. We manually copied and pasted all required open-access review paper sections, and do not distribute any contents of these review papers.

## A.5 EVIDENCEBENCH AND EVIDENCEBENCH-100K DATASETS STRUCTURE

EvidenceBench uses a train, dev, test split. EvidenceBench-100k uses a train and test split. All datasets have the same structure.

Each data instance (stored in a JSON) has the following features:

- `hypothesis`: the biomedical hypothesis in string format.
- `paper_as_candidate_pool`: an ordered tuple of strings. Each string is one sentence from the paper. This serves as the candidate pool for all of the evidence retrieval tasks.
- `aspect_list_ids`: a list of strings. Each string is an id for a study aspect.
- `results_aspect_list_ids`: a list of strings. Each string is an id for an aspect related to the study's results.
- `aspect2sentence_indices`: a mapping (i.e. dictionary) from each aspect to all sentence indices that are sources of information for that aspect.
- `sentence_index2aspects`: a mapping (i.e. dictionary) from each sentence index to all aspects that this sentence is a source of information for.
- `evidence_retrieval_at_optimal_evaluation`: A dictionary that contains information for evaluating a model's performance on the task Evidence Retrieval @Optimal.
  - `optimal`: A positive integer, which is the smallest number of sentences needed to cover all study aspects.
  - `one_selection_of_sentences`: a list of sentence indices, containing the smallest number of sentences needed to cover all aspects. Note, there are potentially other lists of sentences of the same size which cover all aspects.
  - `covered_aspects`: the list of aspects that are covered, which is all aspects in this case.
- `evidence_retrieval_at_10_evaluation`: A dictionary that contains information for evaluating a model's performance on the task Evidence Retrieval @10.
  - `one_selection_of_sentences`: a list of 10 sentence indices. This list covers the maximum number of aspects which can be covered by 10 sentences.
  - `covered_aspects`: the list of aspects that are covered, which may be fewer than all aspects.
- `results_evidence_retrieval_at_optimal_evaluation`: A dictionary that contains the information for evaluating a model's performance on the task Results Evidence Retrieval @Optimal. The structure is similar to `evidence_retrieval_at_optimal_evaluation`.
- `results_evidence_retrieval_at_5_evaluation`: A dictionary that contains the necessary information for evaluating a model's performance on the task Results Evidence Retrieval @5. The structure is similar to `evidence_retrieval_at_10_evaluation`.

- `sentence_types_in_candidate_pool`: a tuple of strings. Each string is a sentence type. There are three possible sentence types: section_name, abstract, and normal_paragraph. For example, if the third string is 'abstract', that means the third sentence comes from the abstract.
- `paper_id`: the id of the paper used as the candidate pool.

# B  ANNOTATION GUIDELINES

## B.1  GUIDELINES FOR HYPOTHESIS VALIDATION

Below, we show the annotation guidelines for evaluating the hypotheses extracted from the review papers. These guidelines were co-designed and approved by a medical doctor who did not see the 50 hypotheses which were evaluated.

**Overall:**

```
IARC is a WHO organization that invites field experts to write
a review about the potential carcinogenicity of a certain
chemical/compound/product/substance, where they survey many
relevant papers.
```

A review is typically organized into the following sections:

- Exposure Data (e.g., how humans and animals come into contact with the substance).
- Animal Study.
- Human Study.
- Mechanistic Evidence (e.g., the mechanism for carcinogenicity).
- Others.

```
For each relevant paper, the field experts will extract certain
information from the paper, for a specific purpose, which does not
have to align with the original goal of the paper.
```

**Annotation Task:**

```
Each task is in a docx file.  In each docx file, you will see:
```

- A hypothesis.
- A paragraph of extracted information from paper.
- A reference page (For reference only).
  - Potentially more context for the hypothesis (i.e., a potential connection between the hypothesis and the extracted information from paper.
  - The review that contains the extracted information from the paper.
  - Link for the paper.

```
You have two tasks.
```

- Determine if the hypothesis is a reasonable hypothesis, given your understanding of the hypothesis and your external knowledge and experience.

```
              - The hypothesis might be about the carcinogenicity of a
                substance (for human or animal), or might be about how
                humans get exposed to a substance, or might be about
                experimental procedure, or something else.
              - Determine if the hypothesis is a valid statement
                with scientific value, it could be a false statement,
                but disproving it would have scientific value.  In
                other words, you should not judge the accuracy of the
                hypothesis.  You should only judge if the hypothesis
                contains scientific value.
              - Determine if the hypothesis looks like a hypothesis,
                i.e., has the format of a real hypothesis.
              - Make your judgment based only on the contents of the
                hypothesis, which is usually just one sentence.  Your
                decision should not be influenced by the other task or
                other materials you see, though for better comprehension,
                you can refer to the links on the reference page.
              - Record your decision(Yes or No), and leave any optional
                comment if you want.  If you think the answer is not
                binary, then you do not have to write yes or no, but you
                have to give an explanation.
        • Determine if the extracted information from the paper
          contains evidence that can potentially help support or
          refute the hypothesis.
              - Answer Yes or No, followed by a brief explanation.  One
                or two sentences.  If you think the answer is not binary,
                then you do not have to write yes or no, but you have to
                give an explanation.
```

**Notes:**

```
You have to pledge the following conditions are met during
annotation for each task packet.

        • No consulting with AI and LLM.

        • For words or concepts that you are not familiar with and
          believe are important for comprehension, search for them and
          understand their meaning.

        • If you do not understand the hypothesis or the extracted
          information from the paper, you should read the IARC review
          to better understand the hypothesis and read the full
          paper to better understand the concepts mentioned in the
          hypothesis and extracted information from the paper.

        • You are not required to read the whole paper nor the
          full IARC review, just to the point when you believe you
          understand the hypothesis and extracted information from the
          paper well enough.
```

### B.2 ANNOTATION GUIDELINES FOR ALIGNMENT OF STUDY ASPECTS AND SENTENCES

#### B.2.1 FIRST ANNOTATION STAGE

This section shows the annotation guidelines for the first stage of aspect-sentence alignment. In this stage, each annotator had to independently annotate 50 papers.

```
You have a total of 50 annotation task packets.  Each task packet
is a docx.  file that contains the following information.
```

- An aspect (one piece of important information/detail).
- The context for the Aspect (summary or a collection of extracted details from a paper).
- The URL for the paper (pmc or pubmed link).
- The list of indexed text elements of the paper (a text element could be a sentence or a section title).

You have to pledge the following conditions are met during annotation for each task packet.

- No consulting with AI or LLM.
- For words you are not familiar with and believe are important for comprehension, conduct a search and understand its meaning.
- Click on the paper URL and find full contents either in HTML, XML, or PDF format, and read through it from start to finish, at least once.
- For every text element in the list, you must look at it and read it at least once.
- You cannot talk to other annotators about anything related to your task, including progress and insights.
- You have to take a mandatory 5-minute break after every 1 hour of performing annotation.
- You cannot exceed 8 hours of annotation per day.

Below is the recommended procedure for annotating each packet.

- Read and understand the aspect, and the context of the aspect.
- Decide what details in the aspect count as information, and what details count as context. At least one detail needs to be identified as information, but an aspect can do without context if it is self-contained and very clear.
  - You can decide what information is, but geographical, temporal, and numerical data are all information.
  - Typically, context is recurring and ubiquitous information throughout the paper.
- For each text element, you decide if a very significant amount of information identified in the aspect is also explicitly present in the text element. Note, information has to be explicitly present, it cannot be from inference or allusion. Acronyms, abbreviations, and different presentation formats of the same information (e.g., rounding of numbers) are acceptable, as long as it is clear to you.
  - Even if you find significant information overlap, you have to make sure the sentence is in the same context as the aspect.
    * Same context typically refers to the same study or experiment.
    * Check if the sentence refers to the same experiment as the aspect, since different experiments could be in one paper.
- Do not do complicated mental inference. Once you have a clear understanding of the aspect and sentence, do not try to invent a spurious connection between sentence and aspect.

```
                    - Specifically, do not do complex computations of numbers.
```

### B.2.2 SECOND ANNOTATION STAGE

Below we show the annotation guidelines for the second stage of the aspect-sentence alignment. In this stage, two annotators from the same team come to a consensus on any disagreements from the first stage.

```
• Go through task 0-49.

• Resolve your difference, check if you made a mistake, or if
  you missed something.  If you made a conceptual error (e.g.
  you failed to understand some terminology), you may have to
  go through the paper again quickly.

    - For sentences that you cannot resolve your difference
      after discussion, i.e., one person says yes, and the
      other person says no, you should include them as well.
```

## C   AUTOMATED ALIGNMENT PROCEDURE

This section describes prompt optimization for the LLM alignment of study aspects and sentences.

Prompt optimization was performed with GPT4-0125 on an independent development set of 37 (aspect, paper) pairs, i.e., 37 tasks. There was no overlap with the papers labeled by the two teams of human annotators.

Given a statement and its context. You are also given a list of indexed text elements from a paper (including tables, figure captions, section titles, sentences, others). Text element is followed by its index, in this format, index: text element [End of text element index].

Focus solely on the statement, go through each text element in the list, determine if it itself alone contains ALL Information in the statement. In other words, you must determine if the text element could be the source of information for the statement. For example, If the statement contains numerical, geographical, and time/date details, its suitable text element must include the same numerical, geographical, and time/date details mentioned in the statement, subject to different formats, such as different rounding, spelling, acronyms.

Statement:
**{aspect}**

Statement context:
**{context}**

Paper's list of text elements:
**{list_of_text_elements}**

Make sure you go through the entire list of text elements, no matter how long. If you did not find any text element, explicitly explain why the information in the statement cannot be found in any text element. For each text element you choose because you think it can be the source of information for the statement,  write to explain why. If you fail to provide such a satisfactory explanation, you should not include that text element. Output all your explanation and reasoning after the keyword "REASON:"

Once you finalize your selection, you should include the numerical index of each text element you choose. Finally, output all chosen numerical index or indices in a python list [index1, index2, ...] after the keyword "DECISION:"

Figure 5: Prompt for aligning a sliding window of consecutive text elements with a study aspect.

In order to reduce the frequency of GPT-4 forgetting information from the papers, we use a sliding window (window-length = 10 sentences) with an overlap of 5 text elements across windows. GPT-4 sees a sliding window of sentences and annotates each sentence according to whether it is a source of information for the aspect. Since the sliding windows are overlapping, each text element (except for the first 5) is considered twice by GPT-4. A sentence is labeled as positive if it is selected in either sliding window.

Figure 5 shows the final prompt template used for aligning text elements with aspects. Each template uses one aspect, 10 text elements in a sliding window, and the context around the one aspect (i.e., the evidence summary of the paper).

## C.1 STUDY ASPECT DECOMPOSITION

There are two steps in aspect decomposition.

**The first step** is decomposing a evidence summary into a list of study aspects where each aspect represents a single piece of information. The granularity of the decomposition is determined by the following rule:

Each decomposed study aspect must be able to align with at least one sentence from the paper. If a study aspect contains so much information that no one sentence can cover a significant portion of these details, then this study aspect is considered too coarse-grained and must be further decomposed. See Figure 6 for the prompt template that achieves the first step of aspect decomposition.

---

You are given an expert statement about a biomedical article **{paper_id}**. Your job is to decompose each sentence into pieces of information.

You are also given all sentences from the original paper as reference. You should determine the level of specificity for the extracted information according to the following criteria: when decomposing the expert statement, make sure that each information point can be covered by at most one sentence from the original paper; if you need more than one sentence to cover the information point, you must decompose that information point further.

Here are some important rules for decomposition:

1. Each piece of extracted information should reveal only one specific aspect about the statement.

2. Do not delete any information from the expert statement.

3. Only output the extracted information.

Here is the expert statement:
**{summary}**

Here are sentences from the original paper:
**{candidate_pool}**

---

Figure 6: Step 1 of decomposing a evidence summary into aspects, using the granularity condition.

**The second step** is checking if the decomposed list of study aspects only contains information from the evidence summary, and if no other paper-specific information leaked into the decomposed list. See Figure 7 for the prompt template. Any datapoint whose decomposed list of aspects did not pass the second step verification is filtered out. Fewer than 10% of datapoints are filtered at this step. Empirically we noticed those filtered datapoints have evidence summaries that are not self-contained. Therefore, we did not attempt to recover these datapoints.

---

Given a statement, and given a decomposed list of aspects for that statement. Your task is to determine if any of the aspects in the list contains any information, details or terminology not explicitly and directly written in the statement. Note, abbreviations and typos are allowed. If you found any aspect that contains information (any information at all), not in the statement, output False after the keyword "Judgment:", otherwise, if you determine all aspects only contain information from the statement, output True after the keyword "Judgment:"

Statement:
**{summary}**

Decomposed list of Aspect:
**{indexed_aspects}**

---

Figure 7: Step 2 of decomposing a evidence summary into aspects. This step confirms that aspects only contain information from the evidence summary.

# D    EXPERIMENT DETAILS

There are several prompt templates used for experimental evaluation, which are variations on a default template.

## D.1    DEFAULT PROMPT TEMPLATE FOR EVIDENCE RETRIEVAL

The default prompt template asks an LLM to retrieve no more than K sentences for the Evidence Retrieval tasks. Figure 8 shows the default prompt template for ER @Optimal or ER @K.

> Given a hypothesis, and a biomedical paper in the format of an indexed list of text elements (including sentences and (sub)section titles), identify the experiment in the paper that provides evidence relevant to the hypothesis. You must explain how this experiment would provide evidence relevant to the hypothesis. Determine what details (i.e. text elements) about the experiment would jointly provide the most effective evidence to support or refute the hypothesis. Your task is to find NO MORE than **{number_of_allowed_text_elements}** text elements. Since you are only allowed to find a limited amount of text elements, you must only select text elements that together would cover the most amount of details of the experiment that you believe are relevant to the hypothesis.
>
> Hypothesis:
> **{hypothesis}**
>
> Paper:
>
> **{indexed_list_of_text_elements}**
>
> Provide an explanation of how the details you identified from the experiment would jointly provide the most effective set of evidence relevant to the hypothesis. Output your reasoning first, and based on your reasoning, make your final selections of text elements. You should only keep the most effective details. Make sure you do not find more than **{number_of_allowed_text_elements}** text elements. If you do, you have to choose the most effective **{number_of_allowed_text_elements}** elements from it that jointly would provide the best evidence relevant to the hypothesis.
>
> Finally, output the list of indices of these text elements in a Python list [index1, index2, ...] after the keyword "DECISION:"

Figure 8: Default prompt template for evaluating LLMs on tasks Evidence Retrieval @Optimal and Evidence Retrieval @10.

## D.2    PROMPT TEMPLATE FOR RESULTS EVIDENCE RETRIEVAL @OPTIMAL OR @5

> Given a hypothesis, and a section of a biomedical paper in the format of an indexed list of text elements (including sentences and (sub)section titles), identify experimental results and analyses that provide evidence which is relevant to the hypothesis. Do not include methods or background. You must explain how the results and analyses provide evidence for or against the hypothesis. Your task is to find NO MORE than **{number_of_allowed_text_elements}** text elements. Since you are only allowed to find a limited amount of text elements, you must determine and select representative text elements that would jointly provide the most effective evidence to support or refute the hypothesis.
>
> As a demonstration, you will first see a sample hypothesis, and a sample list of representative text elements that are about experimental results and analyses, and together cover the most amount of details relevant to the hypothesis.
>
> Sample Hypothesis A, (if you have to find no more than m sentences)
> …
> Sample List A (m sentences):
> [...]
>
> Now you will see the actual hypothesis, and a section of the biomedical paper. Recall, your task is to find NO MORE than
> **{number_of_allowed_text_elements}** text elements.
>
> Hypothesis:
> **{hypothesis}**
>
> A section of the paper as an indexed list:
> **{indexed_list_of_text_elements}**
>
> Note, you must only consider text elements related to experimental results and analyses. Provide an explanation of how the details you identified from the experimental results and analyses would jointly provide the most effective set of evidence relevant to the hypothesis. Output your reasoning first, and based on your reasoning, make your final selections of text elements. Make sure you do not find more than **{number_of_allowed_text_elements}** text elements, if you do, you have to choose the most effective **{number_of_allowed_text_elements}** elements from it that jointly would provide the best evidence relevant to the hypothesis.
>
> Finally, output the list of indices of these text elements in a python list [index1, index2, ...] after the keyword "DECISION:"

Figure 9: Step 1: Prompt template for evaluating LLMs on tasks **Results** Evidence Retrieval @Optimal and Evidence Retrieval @10. Here, the number of allowed text elements denotes Optimal or 10. This prompt uses a mixture of one-shot ICL and Section-by-Section.

**Step 1 of Prompt template with one-shot ICL and Section-by-Section**

Figure 9 shows the prompt template for any tasks that only focus on retrieving sentences related to experiment results or analyses based on experiment outcomes. Note, this prompt uses a mixture of two strategies: one-shot ICL (in-context-learning) and section-by-section processing. These two strategies are proven effective in the other two tasks, Evidence Retrieval @Optimal and @K. Due to budget limitations, we can only provide this mixture strategy, which proves to be the best strategy on the training set. Note, for fairness, for each section, we can only instruct the LLM to retrieve no more than K sentences, even though a paper could have 10 sections. Consequently, the total number of retrieved sentences for all sections combined sometimes exceed to maximally allowed number of sentences K. Therefore, we have the second step of processing.

**Step 2 of Prompt template with one-shot ICL and Section-by-Section:**

In step 2, we show the LLM all its retrieved sentences and ask it to select the top K sentences. See Figure 10 for its prompt template.

---

Given a hypothesis, and an indexed list of text elements (including sentences and (sub)section titles) retrieved by you from a biomedical paper, identify text elements about experimental results and analyses that provide evidence which is relevant to the hypothesis. Do not include methods or background. Find NO MORE than **{number_of_allowed_text_elements}** text elements that would jointly provide the most effective evidence to support or refute the hypothesis.

Hypothesis:
**{hypothesis}**

List of text elements:
**{indexed_list_of_text_elements}**

Note, you must only consider text elements related to experimental results and analyses. Provide an explanation of how the details you identified from the experimental results and analyses would jointly provide the most effective set of evidence relevant to the hypothesis. Output your reasoning first, and based on your reasoning, make your final selections of text elements. Make sure you do not find more than **{number_of_allowed_text_elements}** text elements, if you do, you have to choose the most effective **{number_of_allowed_text_elements}** elements from it that jointly would provide the best evidence relevant to the hypothesis.

Finally, output the list of indices of these text elements in a python list [index1, index2, ...] after the keyword "DECISION:"

---

Figure 10: Step 2: Prompt template for evaluating LLMs on tasks **Results** Evidence Retrieval @Optimal and Evidence Retrieval @10.

## D.3 ICL PROMPT FOR EVIDENCE RETRIEVAL @OPTIMAL AND @10

We randomly selected 8 pairs of examples from the development set, which was completely disjointed from the test set. We experimented with different versions of in-context learning. In one attempt, we gave the full paper for each example pair (i.e., sample hypothesis, sample full paper, sample optimal number of or 10 sentences that cover the most amount of study aspects.). However, no LLM improved on the training set using N-shot with full paper, even when N =1 or 2. Therefore, we decided to not use the full paper. Instead, for each example pair, we give only the sample hypothesis and the sample list of sentences that cover the maximum amount of aspects (size = Optimal or 10). See Figure 11 for its prompt template.

---

Given a hypothesis, and a biomedical paper in the format of an indexed list of text elements (including sentences and (sub)section titles), identify the experiment in the paper that provides evidence relevant to the hypothesis. You must explain how this experiment would provide evidence relevant to the hypothesis. Determine what details (i.e. text elements) about the experiment would jointly provide the most effective evidence to support or refute the hypothesis. Your task is to find NO MORE than **{number_of_allowed_text_elements}** text elements. Since you are only allowed to find a limited amount of text elements, you must only select text elements that together would cover the most amount of details of the experiment that you believe are relevant to the hypothesis.

As a demonstration, you will first see 8 pairs of examples, each example will have a sample hypothesis, and a sample list of text elements that are representative and together cover the most amount of details of a relevant experiment.

Sample Hypothesis A, (if you have to find no more than m sentences)
…
Sample List A (m sentences):
[...]
Sample Hypothesis B, (if you have to find no more than n sentences)
…
Sample List B (n sentences):
[...]

…
Sample Hypothesis H, (if you have to find no more than p sentences)
…
Sample List H (p sentences):
[...]

Now you will see the actual hypothesis, and the entire paper. Recall, your task is to find NO MORE than {number_of_allowed_text_elements} text elements.

Hypothesis:
**{hypothesis}**

Paper:

**{indexed_list_of_text_elements}**

Provide an explanation of how the details you identified from the experiment would jointly provide the most effective set of evidence relevant to the hypothesis. Output your reasoning first, and based on your reasoning, make your final selections of text elements. You should only keep the most effective details. Make sure you do not find more than **{number_of_allowed_text_elements}** text elements, if you do, you have to choose the most effective **{number_of_allowed_text_elements}** elements from it that jointly would provide the best evidence relevant to the hypothesis.

Finally, output the list of indices of these text elements in a python list [index1, index2, ...] after the keyword "DECISION:"

---

Figure 11: 8-shot In-context Learning Prompt template for evaluating LLMs on tasks Evidence Retrieval @Optimal and Evidence Retrieval @10.

## D.4 SECTION-BY-SECTION PROMPT FOR EVIDENCE RETRIEVAL @OPTIMAL AND @10

Section-by-section is a strategy to counter the long-context difficulty posed by EvidenceBench. Instead of processing the full paper at once (typically consisting of more than 5000 tokens), each paper is divided into its naturally defined sections, i.e. introduction, methodology, results, etc. Each time, an LLM only retrieves sentences from one single section. Note, for fairness, for each section, we can only instruct the LLM to retrieve no more than K sentences, even though a paper could have 10 sections. Consequently, the total number of retrieved sentences for all sections combined sometimes exceeds the maximally allowed number of sentences K. Therefore, we have the second step of processing where we ask the model to select the best K sentences from all sentences retrieved from all sections. See Figure 12 for the prompt template of the first step. See Figure 13 for the prompt template that asks the LLM to choose the best K sentences from all its retrieved sentences from all sections.

---

Given a hypothesis, and a section from biomedical paper in the format of an indexed list of text elements (including sentences and (sub)section titles), identify the experiment in the section that provides evidence relevant to the hypothesis. You must explain how this experiment would provide evidence relevant to the hypothesis. Determine what details (i.e. text elements) about the experiment would jointly provide the most effective evidence to support or refute the hypothesis. Your task is to find NO MORE than **{number_of_allowed_text_elements}** text elements. Since you are only allowed to find a limited amount of text elements, you must only select text elements that together would cover the most amount of details of the experiment that you believe are relevant to the hypothesis.

Hypothesis:
**{hypothesis}**

Paper:

**{indexed_list_of_text_elements}**

Provide an explanation of how the details you identified from the experiment would jointly provide the most effective set of evidence relevant to the hypothesis. Output your reasoning first, and based on your reasoning, make your final selections of text elements. You should only keep the most effective details. Make sure you do not find more than **{number_of_allowed_text_elements}** text elements, if you do, you have to choose the most effective **{number_of_allowed_text_elements}** elements from it that jointly would provide the best evidence relevant to the hypothesis.

Finally, output the list of indices of these text elements in a python list [index1, index2, ...] after the keyword "DECISION:"

---

Figure 12: Step 1: Section-by-section Prompt template for evaluating LLMs on tasks Evidence Retrieval @Optimal and Evidence Retrieval @10.

---

Given a hypothesis, and an indexed list of text elements (including sentences and (sub)section titles) retrieved by you from a biomedical paper, find NO MORE than **{number_of_allowed_text_elements}** text elements that would jointly provide details of an experiment to support or refute the hypothesis.

Hypothesis:
**{hypothesis}**

List of text elements:

**{indexed_list_of_text_elements}**

Explain how the text elements would collectively describe an experiment to support or refute the hypothesis. Output your reasoning first, after the keyword "REASON:", then make your final selections of text elements. Make sure you do not find more than **{number_of_allowed_text_elements}** text elements, if you do, you have to choose the most effective **{number_of_allowed_text_elements}** elements from it that would most effectively support or refute the hypothesis.

Finally, output the list of indices of these text elements in a python list [index1, index2, ...] after the key word "DECISION:"

---

Figure 13: Step 2: Section-by-section Prompt template for evaluating LLMs on tasks Evidence Retrieval @Optimal and Evidence Retrieval @10. The LLM is asked to choose the best K number of sentences.

## D.5 REGENERATION PROMPT

If an LLM retrieved more than allowed despite explicit instructions, it will be sent a regeneration prompt, with the following format in Figure 14.

---

**{Previous conversation}**

{'role': 'user', 'content': You have output more than **{number_of_allowed_text_elements}** text elements, which violates the requirement to find no more than **{number_of_allowed_text_elements}** number of text elements. You must choose the best **{number_of_allowed_text_elements}** text elements in the same format as the previous output.}

---

Figure 14: Regeneration prompt template if LLM exceeds the maximally allowed number of text elements.

## D.6 INSTRUCTIONS FOR EMBEDDING MODEL

For Evidence Retrieval @Optimal and @10, for embedding models, there are two strategies, with instruction or without instruction. The instruction is:

```
"From a biomedical experiment, find important and representative
details that would form the most effective set of evidence
relevant to the hypothesis"
```

Recall, "with instruction strategy" means concatenating the instruction with the hypothesis and creating an embedding for the concatenated text as a new hypothesis vector. Each text element in the candidate pool (i.e. the paper) is still independently embedded as its own vector.

See Table 6 for embedding models' performance with instruction and without instruction, on the tasks of Evidence Retrieval @Optimal and @10.

Table 6: Embedding Models Comparison. Standard Error calculated by bootstrapping.

| | No Instruction | | Instruction | |
|---|---|---|---|---|
| **Model** | ER @ optimal | ER @ 10 | ER @ optimal | ER @ 10 |
| BM25 | $16.5 \pm 1.1$ | $34.4 \pm 1.7$ | - | - |
| OpenAI | $\mathbf{25.1 \pm 1.4}$ | $\mathbf{42.2 \pm 1.7}$ | $23.8 \pm 1.4$ | $41.3 \pm 1.7$ |
| VoyageAI | $21.6 \pm 1.3$ | $42.0 \pm 1.8$ | $22.7 \pm 1.3$ | $41.9 \pm 1.8$ |
| GritLM | $21.5 \pm 1.2$ | $39.7 \pm 1.7$ | $\mathbf{27.0 \pm 1.3}$ | $\mathbf{46.4 \pm 1.7}$ |
| E5-Mistral | $22.7 \pm 1.4$ | $41.9 \pm 1.8$ | $21.9 \pm 1.3$ | $40.8 \pm 1.8$ |

For tasks such as Results Evidence Retrieval @Optimal and @10, embedding models will take in task-specific instruction. Instruction has the following format:

```
From a biomedical paper, find important and representative details
about experiment outcomes, results and analyses that would form
the most effective set of evidence relevant to the hypothesis.
```

Note, in the instruction, we have concisely and explicitly informed the embedding model that the embedding of the hypothesis should only have high cosine similarity with text elements related to results or analyses. Since we are not changing the embedding for the results and analyses related text elements, we only change the embedding for the hypothesis to fit this purpose. This is an emergent and specially fine-tuned ability of some of the newer and more powerful embedding models, such as GritLM. See Table 7 lower right quadrant for the performance of embedding models with instruction on tasks such as Results Evidence Retrieval (ER) @Optimal and @5.

## D.7  Standard Errors for Model Evaluations

In this section, we reproduce the main results from the paper, showing standard errors for all estimates.

In table 7, we show overall results for model performance on the 4 tasks. For Evidence Retrieval Tasks, three strategies are considered: default, in-context learning, and section-by-section for each LLM. Default refers to Figure 8 prompt. ICL refers to Figure 11. Section-by-Section refers to Figure 12 and 13. For each model, the strategy that achieved the best performance is selected, and that result is reported as the model's performance. For example, for the task ER @optimal, gpt-4o achieves the best aspect recall using ICL, whereas for the task ER @10, gpt-4o achieves the best aspect recall using section-by-section. Note, for embedding models, there are two strategies: with instruction or without instruction. The best performance for each embedding model is recorded.

Note, for the two tasks Results Evidence Retrieval (ER) @Optimal and @5, only one strategy is considered for LLM, which is one-shot ICL with section-by-section processing, see Figure 9 and Figure 10.

For Results Evidence Retrieval (ER) tasks, embedding models must only have one strategy, i.e. the strategy with instruction. Since the task is about retrieving sentences related to results and analyses, an embedding model must understand this constraint through its instructions.

Table 7: Aspect Recall for the full retrieval task (top) and the result retrieval task (bottom). For each model, the highest number is reported if multiple strategies are used. Standard Error calculated by bootstrapping.

| | Max | Random | GPT-4o | Claude3-Opus | Gemini-1.5 | LLaMA3-70b | OpenAI Emb | VoyageAI | GritLM | E5-Mistral |
|---|---|---|---|---|---|---|---|---|---|---|
| ER @ optimal | 100.0 | 9.6 | $51.4 \pm 1.4$ | $47.6 \pm 1.5$ | $48.3 \pm 1.4$ | $46.7 \pm 1.4$ | $25.1 \pm 1.4$ | $22.7 \pm 1.3$ | $27.0 \pm 1.3$ | $22.7 \pm 1.4$ |
| ER @ 10 | 99.3 | 22.3 | $71.6 \pm 1.5$ | $66.4 \pm 1.6$ | $65.4 \pm 1.6$ | $65.4 \pm 1.6$ | $42.2 \pm 1.7$ | $42.0 \pm 1.8$ | $46.4 \pm 1.7$ | $41.9 \pm 1.8$ |
| Result ER @ optimal | 100.0 | 4.4 | $52.6 \pm 2.1$ | $51.7 \pm 2.1$ | $46.7 \pm 2.0$ | $46.2 \pm 2.2$ | $19.1 \pm 1.8$ | $18.3 \pm 1.8$ | $18.9 \pm 1.7$ | $19.3 \pm 1.8$ |
| Result ER @ 5 | 99.8 | 11.4 | $70.8 \pm 1.9$ | $68.7 \pm 2.0$ | $65.4 \pm 2.0$ | $63.7 \pm 2.1$ | $33.1 \pm 2.2$ | $31.9 \pm 2.2$ | $39.1 \pm 2.2$ | $33.6 \pm 2.2$ |

Table 8: Comparison of prompting strategies for LLMs. Baseline refers to Figure 8 prompt. ICL refers to Figure 11. Sec-by-Sec refers to Figure 12 and 13. Standard errors are calculated by bootstrapping.

| | Baseline | | ICL | | Sec-by-Sec | |
|---|---|---|---|---|---|---|
| **Model** | ER @ optimal | ER@ 10 | ER @ optimal | ER@ 10 | ER @ optimal | ER@ 10 |
| GPT-4o | $\mathbf{48.1 \pm 1.5}$ | $\mathbf{69.6 \pm 1.5}$ | $51.4 \pm 1.4$ | $68.7 \pm 1.6$ | $50.9 \pm 1.4$ | $\mathbf{71.6 \pm 1.5}$ |
| Claude3-opus | $41.1 \pm 1.6$ | $53.6 \pm 1.6$ | $38.3 \pm 1.4$ | $55.4 \pm 1.7$ | $47.6 \pm 1.5$ | $66.4 \pm 1.6$ |
| Gemini-1.5 | $42.7 \pm 1.5$ | $63.0 \pm 1.6$ | $43.2 \pm 1.5$ | $62.4 \pm 1.7$ | $48.3 \pm 1.4$ | $65.4 \pm 1.6$ |
| LLaMA3-70b | - | - | - | - | $46.7 \pm 1.4$ | $65.4 \pm 1.6$ |

Table 8 shows model performance for the three prompting strategies.

We observe that all standard errors are less than 2%, indicating we have a sufficiently large test set size to effectively distinguish different models and prompting strategies.

## E    MODEL SENSITIVITY TO PARAPHRASED HYPOTHESIS

We test models' sensitivity to different paraphrased versions of the hypothesis. We use a variety of LLMs to paraphrase a hypothesis. Note, a paraphrase would still keep the scientific terminology to make sure it is still the same hypothesis. As shown in Figure 9, GPT-4o and the two embedding models are less sensitive to paraphrased hypotheses, while Claude3-Opus is more sensitive to them.

Table 9: Models evaluated under paraphrased versions of the hypothesis. All experiments are under task Results ER @ optimal.

| Model | Original Hypothesis | GPT-4o Paraphrased | Claude Paraphrased | Llama3-70B Paraphrased | Llama3-8B Paraphrased |
|---|---|---|---|---|---|
| GPT-4o | 52.6 | 49.9 | 51.7 | 50.8 | 50.3 |
| Claude3-Opus | 51.7 | 44.6 | 41.9 | 43.3 | 43.9 |
| GritLM | 18.9 | 22.8 | 19.4 | 20.8 | 19.1 |
| OpenAI Emb | 19.1 | 19.1 | 18.1 | 19.7 | 18.3 |

# F HYPOTHESIS COLLECTION

There are two steps for collecting hypotheses from the review papers.
**Step 1**:
See Figure 15 for the prompt template. Note that Claude3-Opus is used for this step. The model is instructed to answer, "What is the motivation for the expert reviewer to extract these pieces of information from the paper and summarize them?".

> You will be provided excerpts from a biomedical monograph. Some of these excerpts will serve as context. There will also be a highlighted excerpt, which was written by domain experts after they read a specific paper.
>
> The highlighted text describes specific pieces of evidence that were extracted from the paper. They may describe experimental background, setup, procedure, methodology, results, and analyses.
>
> The domain experts extracted these specific pieces of evidence from the paper because they were trying to provide evidence for a certain hypothesis. Your task is to identify this hypothesis.
>
> In performing this task, you should think about the reasons why the domain expert extracted these specific pieces of evidence. Which hypothesis is best supported by these pieces of evidence? The hypothesis should be specifically tailored for these pieces of evidence from the highlighted text.
>
> The hypothesis should not include superficial details from the highlighted excerpt, i.e. it should not be a mere reformulation of the evidence. The hypothesis should be related to the context surrounding the highlighted excerpt, and cannot semantically deviate from the surrounding context. You should write the hypothesis mainly using terminology, phrases and sentences from the surrounding context. Do not use wording from the highlighted excerpt itself. The hypothesis should not discuss limitations of the study.
>
> **Excerpts from a biomedical monograph**:
> **{context}**
> **The Highlighted Excerpt**:
> **{paper_summary}**
>
> Think step by step before writing the final version of the hypothesis by taking into account all above requirements. As a reminder, the hypothesis cannot use details from the highlighted excerpt, the hypothesis must arise from the surrounding context, and the hypothesis must be fully supported by the entirety of the highlighted excerpt. The hypothesis should not discuss limitations of the study.
>
> Finally, output the hypothesis after the keyword "HYPOTHESIS:"

Figure 15: First step of hypothesis collection from review papers

**Step 2:**
In order to make sure the hypotheses have the correct format and do not have any summarized evidence, we perform a second step where we trim the first-step output off any unnecessary details and specifications and only preserve the central biomedical question. See Figure 16 for the prompt template.

> Consider a paragraph from a biomedical monograph:
> **{context_paragraph}**
>
> And Consider the following claim:
> **{longer_hypothesis}**
>
> Using the paragraph as context, what is the main hypothesis being stated in the claim? Disregard any summarized evidence from the claim that does not fit into the central hypothesis. Importantly, if the claim contains a speculative question, keep the part of the claim that contains the speculative question. Think step by step, and output your reasoning. After you are done reasoning, you should have identified the central component of the hypothesis (with respect to the paragraph from the biomedical monograph), and the speculative question, if the claim has such a question. Explicitly state the central component, then make the speculative question definitive. Incorporate it into the central component, and transform it into a natural biomedical hypothesis. The transformed hypothesis should be clear, concise, definitive and unambiguous. Remember, the transformed hypothesis cannot have any detailed or summarized evidence from the claim, make sure to check this!
> After you have transformed the hypothesis, remember to check that you DID NOT ADD any information that is not explicitly stated in the claim. Also remember to check no summarized evidence or details from the claim in any form is included in the hypothesis. Output the central hypothesis after the keyword "HYPOTHESIS:"

Figure 16: Second step of hypothesis collection, trimming the output of step 1.

# G    DATA PREPROCESSING

## G.1    HARVESTING EVIDENCE SUMMARY

We extract expert-written evidence summaries from review papers using an algorithmic procedure. Summaries that cite multiple papers are excluded. We also exclude any paper that is not licensed under Creative Commons or is not in the public domain. Finally, we remove any paper that is cited twice in one section of a review paper, ensuring that the extracted evidence summaries are complete.

## G.2    FURTHER PROCESSING EVIDENCE SUMMARY

The primary difficulty in ensuring the high quality of evidence summary extracted by the algorithm lies in the variability in the placement of the citation (e.g., Hoang et al. (2016)). Since review papers sometimes have hundreds of papers cited in one section, each with a evidence summary surrounding the citation, it is challenging to heuristically delineate the boundaries of the evidence summaries, and motivates the use of an LLM for this step. See Figure 17 for prompt template

> Given a paragraph in a biomedical report.
>
> Paragraph:**{paper_summary}**
>
> Determine what sentence(s) describe experimental details and observations
> from this particular paper: **{paper_url}**. Must include the sentence that contains the paper id **{paper_url}**, do not remove the paper id.
> DO NOT include summaries, comments, opinions or limitations about this specific paper. Limitations are usually enclosed by [] and might include phrases such as "the working group".
>
> Output the selected sentence(s) VERBATIM after the keyword HINT:

Figure 17: Prompte for extracting evidence summary for a cited paper

## G.3    IDENTIFYING SUITABLE EVIDENCE SUMMARIES

An evidence summary that is suitable for EvidenceBench contains experimental outcomes, results, analyses, or methodology. However, some evidence summaries only have high-level information about a paper and do not have the desired level of specificity. We use an LLM to determine if a evidence summary suits EvidenceBench. See Figure 18 for prompt template.

> Statement:
> **{paper_summary}**
>
> The above statement is from the paper it cited. The text content in the statement is regarded as evidence from the paper.
> Do any part of this evidence describe experimental designs, setup, procedures, methodologies?
> or Do any part of these describe results, findings and conclusions?
> or Does this evidence describe both experimental details as well as results/findings?
> or if these evidence neither describe experimental details nor describe results/findings.
>
> Simply and only output your decision by choosing one of the 4 categories (experiment, results, both, neither), output your chosen category after the keyword "DECISION":

Figure 18: Prompt template to determine if a evidence summary is suitable for EvidenceBench. Here, "both" or "results" is acceptable.

## G.4    HUMAN VERIFICATION

We randomly sampled 50 extracted evidence summary using the entire harvesting procedure and LLM preprocessing. Only in 1 case did we notice where the extracted evidence summary includes a sentence that does not belong to this specific paper. This ensures the high quality and accuracy of the harvesting procedure and LLM preprocessing.

## H    Fine-tuning and Evaluation on EvidenceBench-100k

EvidenceBench-100k also split into a 80k train set and a 20k test set. For cost reasons, we random sample 300 datapoints from the 20k test set. We evaluated various representative LLMs and embedding models on the Task of Result ER@Optimal on this new 300 points test set from EvidenceBench-100k. From table 5a, we clearly see LLMs dominate over embedding models, while GritLM-7B outperforms other embedding models. This shows EvidenceBench-100k test set can also be used to evaluate and differentiate various models' performance.

Furthermore, to demonstrate the quality of EvidenceBench-100k's training set, we fine-tuned two models. E5-v2 335M is a light-weight embedding model. Since there are over 100 million sentence-aspect pair judgments from the train set, we randomly sampled 1 million triplets, where each triplet contains an anchor (the study aspect), a positive (a sentence that is considered as source of information for the study aspect), a negative (a sentence that is not considered as source of information for the aspect). Using a margin-based triplet-loss with margin=0.05, AdamW optimizer (weight decay =0.01), peak learning rate of 5e-4, 10% linear warmup then cosine-annealing, batch size =256, we trained E5-v2 with full-parameter tuning for one epoch. We also fine-tuned Llama3-8B using all 80k training datapoints. Llama3-8B is trained with LoRA rank=8, alpha =16, a batch size of 16, AdamW optimizer (weight decay =0.01), and the same learning rate scheduler, we fine-tuned the model for one epoch. Note, during inference time we trained LLama3-8B to only output sentence indices, during training, we trained it to output both sentence indices, sentence texts and the summarized list of study aspects. This has shown to be more effective than only training it on sentence indices.

We made sure none of the papers in the original EvidenceBench is used in any way in the EvidenceBench-100k to avoid contamination. Our fine-tuned E5 and Llama3-8B are tested on the original EvidenceBench test set for the task of Result ER@Optimal.

Notice that the fine-tuned E5 model has dramatically improved its performance on the original test set, surpassing all larger embedding models, see Table 5b. It only trails behind the performance of large language models. The fine-tuned Llama3-8B has a performance of 41.0% which trails behind much larger SoTA LLMs. This shows that EvidenceBench-100k is suitable for developing and training both embedding-based information retrieval systems as well as large language models.

## I    Qualitative Analysis for GPT-4o on the Original EvidenceBench

For the task Result ER@ Optimal, there are a total of 1228 results study aspects that the review papers identified.

Out of these 1228, GPT-4o's retrieved sentences fail to cover 583 study aspects, from which 50 pairs (missed "Result" aspect, set of GPT-4o retrieved sentences) are randomly sampled and manually inspected by two researchers. Out of the 50 cases, 10 cases should not be considered GPT-4o errors. In 7 cases, GPT-4o retrieved an almost sufficient set of sentences but missed one aspect due to the upper limit on the number of sentences. This issue arose from a failure in the optimization step during retrieval, not from a reasoning or retrieval error. In 1 case, our alignment annotation procedure missed one of the GPT-4o retrieved sentences as the source of information for a study aspect. In 1 case, a parsing error occurred where our algorithm did not extract the full summary from the review paper. Finally, in 1 case, one method-related study aspect is mistakenly labeled as a result-related study aspect.

## J   AUTHOR STATEMENT

The authors affirm that they are the sole creators of the submitted manuscript and accept all responsibility for its contents. They warrant that this work is original, has not been published elsewhere, and is not currently under consideration for publication elsewhere. In the event of any violation of rights, the authors bear all responsibilities. Furthermore, the authors confirm that all data used in this research complies with the necessary data licensing requirements.

