# OpenReview forum: "EvidenceBench: A Benchmark for Extracting Evidence from Biomedical Papers"
_ICLR.cc/2025/Conference — Submitted to ICLR 2025_

### Official Review · Reviewer_FodM · 2024-10-28

**Soundness:** 3
**Presentation:** 3
**Contribution:** 2
**Rating:** 5
**Confidence:** 3

**Summary:**

This paper proposes a new task of finding evidence for a hypothesis. The authors built a large-scale dataset with reasonable costs using existing survey monographs and LLMs.

**Strengths:**

- The large-scale datasets are constructed using LLMs with fewer budgets.
- Several evaluations of existing and fine-tuned models are provided and compared, showing the usefulness of the benchmark dataset for evaluating current LLMs' abilities.

**Weaknesses:**

- It needs to be clarified how the hypotheses from survey monographs are generally helpful.
- The authors expect to provide immediate value to scientists as the first desiderate of the benchmark (lines 44-45), and the dataset creation involves several experts. Still, no manual analyses are provided for the results, so whether the results benefit scientists is unclear.
- The proposed task focuses on the limited part of the practical problem; the task expects the candidate pool and needs to consider cases with evidence and with (the retrieval results of) the considerable paper pool (e.g., the entire PubMed database).

Typo:
- Line 509, Figure 3 should be Table 6.

**Questions:**

- Hypotheses are taken from survey monographs, so they can differ from natural hypotheses the experts consider. Is there any evaluation of the dataset that evaluates the suitableness of the instances for evidence retrieval?
- What happens if the experts think of the hypotheses with no evidence?

---

> ### Author Response · Authors · 2024-11-20
> **Rebuttal for Reviewer FodM**
>
> All line numbers, sections, figures and tables refer to our revised version of the paper.  (Note that “study monographs” are now referred to as “review papers.”)
>
>
> >“It needs to be clarified how the hypotheses from survey monographs are generally helpful.”
>
> The hypotheses in our dataset are directly extracted from review papers (such as survey monographs), where they represent real scientific questions that medical experts deemed important enough to warrant comprehensive review. As validated by our expert panel (Section 3.3.1), these hypotheses have clear scientific value and reflect genuine research priorities in biomedicine (line 272-275). This grounding in actual review papers, combined with expert validation, is a key strength of our work.
>
> >“The authors expect to provide immediate value to scientists as the first desiderate of the benchmark (lines 44-45), and the dataset creation involves several experts. Still, no manual analyses are provided for the results, so whether the results benefit scientists is unclear. ”
>
> We have revised our introduction to be more precise about our contribution.
>
> Our work addresses a crucial step in systematic review creation (line 45-52) - a process valued highly enough that research organizations invest millions of dollars annually in systematic reviews. Our benchmark evaluates models' ability to automate evidence extraction, a key bottleneck in this process.
>
> The dataset makes three contributions to the scientific community. First, a well-defined task that addresses a concrete need in systematic review creation. Second, high-quality benchmarks suitable for evaluating and training language models. Third, statistically rigorous pipeline for synthetic dataset generation that can guide future work.
>
> If you have specific manual analyses in mind that would further demonstrate value to scientists, we welcome your suggestions.
>
>
> >“The proposed task focuses on the limited part of the practical problem; the task expects the candidate pool and needs to consider cases with evidence and with (the retrieval results of) the considerable paper pool (e.g., the entire PubMed database).”
>
> Our benchmark focuses specifically on evidence extraction from papers, not document-level retrieval. As clarified in our revised introduction, while established tools and practices exist for searching structured databases like MEDLINE for relevant papers (line 46-47), automatically extracting relevant evidence from these papers remains a key challenge in systematic review creation. This evidence extraction step is precisely what EvidenceBench is designed to evaluate.
>
>
> >Typo:Line 509, Figure 3 should be Table 6.
>
> Fixed. Now changed to Table 5(b). We moved some details of fine-tuning to Appendix H.
>
> >“Hypotheses are taken from survey monographs, so they can differ from natural hypotheses the experts consider. Is there any evaluation of the dataset that evaluates the suitableness of the instances for evidence retrieval?”
>
> Our expert validation process in Section 3.3.1 addresses this concern in two ways. First, experts confirmed our hypotheses have sufficient scientific value. Second, experts verified that the evidence summaries provide relevant evidence for these hypotheses.
>
> The suitability of our dataset instances (sentences) for evidence retrieval is established through a chain of validation. Our annotation procedure correctly identifies sentences that cover study aspects. These study aspects come from expert-written evidence summaries. Finally, experts confirmed these summaries provide evidence related to the hypotheses.
>
> While our hypotheses may differ from those initially considered by the review authors, this does not affect dataset quality - what matters is that they are scientifically valuable and that there is evidence which is relevant for them, both of which are validated by experts.
>
>
> >What happens if the experts think of the hypotheses with no evidence?
>
> Our pipeline works with review papers that contain evidence summaries written by experts. If experts investigated a hypothesis and found evidence against it (for example, a drug having no effect, or a proposed mechanism being disproved), this negative evidence would be captured in their evidence summary and included in our dataset. Our pipeline treats evidence that disproves a hypothesis as equally valid and important as evidence that supports it - both types help answer the scientific question posed by the hypothesis.

---

> > ### Author Response · Authors · 2024-11-23
> > **Looking forward to your reply**
> >
> > Dear Reviewer FodM,
> >
> > Thank you for your valuable feedback. We have rewritten our introduction to explain the real-world motivation for our task. As we are approaching the rebuttal deadline, we would greatly appreciate your thoughts on whether our revisions have adequately addressed your concerns. If any aspects still require clarification, we are happy to provide additional details promptly.

---

> > > ### Author Response · Authors · 2024-11-24
> > > **Looking forward to your reply**
> > >
> > > We deeply appreciate your dedication in reviewing our manuscript. With the rebuttal deadline drawing near, we would be grateful if you could examine our updated paper that reflects your valuable suggestions. We await your feedback.

---

> > > > ### Comment · Reviewer_FodM · 2024-11-26
> > > >
> > > > Thank you for the comments and updates. I would like to keep my original score.

---

### Official Review · Reviewer_YWQR · 2024-10-28

**Soundness:** 3
**Presentation:** 3
**Contribution:** 3
**Rating:** 6
**Confidence:** 4

**Summary:**

This paper introduces EvidenceBench, a scalable annotation pipeline designed for extracting and aligning evidence with specific hypotheses in biomedical literature. Study aspects and hypotheses are initially extracted from systematic reviews. A specialized alignment annotator then performs sentence-level annotations to link each piece of evidence directly to the corresponding hypothesis. To validate this approach, the authors use the pipeline to generate the expansive EvidenceBench-100k benchmark. Fine-tuned on this benchmark dataset, embedding models showed improved performance in the 'Result ER@Optimal' task, showing this standardized benchmark and evaluation framework will support the development of tools for automate evidence synthesis and hypothesis testing.

**Strengths:**

o	The original EvidenceBench evaluated the performance of selected LLMs and embedding models on evidence retrieval tasks and compared different prompting strategies, which revealed that GPT-4o is the SOTA LLM, and that embedding models underperform due to a lack of context awareness.

o	The EvidenceBench-100k fine-tuned E5-v2 model and Llama3-8B significantly improved on the result evidence retrieval task but trailed behind larger models, validating the effectiveness of the benchmark dataset.

o	The author presented the topic and their framework well, with detailed descriptions and clear figures illustrating the overall problem and their area.

o	This paper innovatively developed the pipeline for evidence retrieval for a given hypothesis and further annotated biomedical papers at the sentence level for better meta-analysis.

o	The authors conducted comprehensive experiments with both open-sourced and closed-sourced LLMs, and a small language model. Human experts were involved to validate study aspects generation and automate sentence annotation to enhance trustworthiness.

**Weaknesses:**

o	The authors claimed that their method using the SOTA LLMs reduces construction time from over 3,000 human hours for EvidenceBench to 3 hours. However, there is no evidence provided regarding how the 3 hours were concluded. Additionally, GPT4-0125, GPT4-o-mini, and Claude3-Opus are used during data generation without explanations of when to choose which.

o	No ablation study regarding how topics can influence the study aspect and hypothesis extraction is provided. The subsets used for experiments are randomly selected without considering the distribution of topics.

o	The experiments are conducted on a subset of the test dataset. Although EventBench-100K is comprehensive and large, only 300 data points are used to evaluate LLMs, which is a very small portion.

**Questions:**

o	How is the 3 human hour for construction using this pipeline concluded? Why are different LLMs (GPT4-0125, GPT4-o-mini, and Claude3-Opus) selected as the tool at different stage of benchmark data creation?

o	The summary extraction from study monographs is performed by LLMs, without human inspection. The summaries are used as input to LLMs for the recovery of hypotheses and decomposition of aspects. Although the aspect decomposition is inspected by humans, how is the summary extraction validated to avoid error propagation?

o	In Task Definition, both versions of the task define the desired sentences as evidence for or against a hypothesis. It is difficult to discern whether sentences classified as counter-evidence in relation to one hypothesis might be more appropriately considered as supportive of an alternative hypothesis. Given the structured nature of the tasks, are there any experiments or human validations conducted to address and clarify these categorizations?

**Details Of Ethics Concerns:**

This paper released a new dataset for evidence extraction from biomedical papers, which requires some ethic evaluations of the dataset creation.

---

> ### Author Response · Authors · 2024-11-20
> **Rebuttal for Reviewer YWQR**
>
> All line numbers, sections, figures and tables refer to our revised version of the paper.
>
>
> >The authors claimed that their method using the SOTA LLMs reduces construction time from over 3,000 human hours for EvidenceBench to 3 hours. However, there is no evidence provided regarding how the 3 hours were concluded. Additionally, GPT4-0125, GPT4-o-mini, and Claude3-Opus are used during data generation without explanations of when to choose which.
>
> >“How is the 3 human hour for construction using this pipeline concluded? Why are different LLMs (GPT4-0125, GPT4-o-mini, and Claude3-Opus) selected as the tool at different stage of benchmark data creation?”
> To clarify, the 3 hours are the time used to call OpenAI and Anthropic API for hypothesis generation and sentence annotation etc. The 3 hours are API time. This is the actual time we used to call GPT4 and Claude3, and it is not an estimate. (line 65-66)
>
> The 3000 human hours are estimated based on how long it takes for a team of two Biomedicine PhD students to manually annotate one paper.
>
> Here is the precise rationale behind this estimation: It takes 20 minutes for a human annotator to align one aspect with a paper. On average, there are 10 aspects per paper. To minimize human error due to factors such as fatigue, two human annotators are required to double-check each other's work. For one paper, it takes 2 * 1/3 * 10 = 20/3 hours to fully annotate all its sentences with the set of aspects. There are 426 papers in the original test set, so it would take 426 * 20/3 = 2840 ≈ 3000 human hours to create EvidenceBench. We assume an hourly wage of 40 to 50 dollars for PhD students, which is half of what is offered in the GPQA Benchmark (https://arxiv.org/abs/2311.12022), where PhD-level annotation is also required. The estimated human cost is 3000 hours, resulting in a total cost of 3000 * 40 = 120,000 dollars
>
> GPT4-0125 was the state-of-the-art at time of dataset construction, so it is used for aspect decomposition, study aspect and sentence annotation. Claude3-Opus is used for hypothesis generation because we observed that it is better at generating more accurate and scientifically valuable hypotheses, as assessed by our expert panels. GPT4o-mini is used for the large-scale construction of EvidenceBench-100k (both the aspect decomposition and sentence annotation) because we validated its performance using our hypothesis testing framework, and because it is 133 times cheaper than GPT4-0125. (0.075 dollars per 1M tokens for GPT4o-mini and 10 dollars per 1M tokens for GPT4-0125). See line 66-69.
>
>
> >No ablation study regarding how topics can influence the study aspect and hypothesis extraction is provided. The subsets used for experiments are randomly selected without considering the distribution of topics.
>
> Random sampling provides an unbiased estimate of the topic distribution in biomedical literature. More importantly, biomedical systematic reviews follow standardized guidelines that are topic-agnostic - the same structured formats and procedures apply across all biomedical domains. During our extensive dataset construction and validation process, we found no evidence that topic differences affect the quality of hypothesis generation or study aspect annotation.
>
> We welcome suggestions for analyses that would be informative for evaluating our pipeline's robustness across different biomedical domains.
>
> >The experiments are conducted on a subset of the test dataset. Although EventBench-100K is comprehensive and large, only 300 data points are used to evaluate LLMs, which is a very small portion.
>
> We originally used 300 datapoints for evaluation due to cost constraints. To address your concern, we evaluated models on an additional 3,000 randomly sampled datapoints (line 430-431). The updated results, shown in Table 5a, are consistent with our findings from the 300-datapoint evaluation, confirming the robustness of our conclusions. For completeness, we provide the original and the updated tables below:
> ### Original table (300 datapoints):
> | Model             | Result ER@Optimal |
> |-------------------|-------------------|
> | GPT-4o            | 40.58%            |
> | Claude3-Opus      | 36.39%            |
> | GritLM            | 14.72%            |
> | OpenAI Embedding  | 9.53%             |
> | E5-Mistral        | 8.20%             |
>
>
> ### Updated table (3000 datapoints):
> | Model      | Result ER@Optimal |
> |------------|-------------------|
> | GPT-4o     | 42.84%        |
> | Claude3-Opus    | 35.12%            |
> | GritLM     | 14.59%            |
> | OpenAI Embedding     | 10.95%            |
> | E5-Mistral | 9.10%             |

---

> > ### Author Response · Authors · 2024-11-20
> > **Continued Author Rebuttal**
> >
> > >The summary extraction from study monographs is performed by LLMs, without human inspection. The summaries are used as input to LLMs for the recovery of hypotheses and decomposition of aspects. Although the aspect decomposition is inspected by humans, how is the summary extraction validated to avoid error propagation?
> >
> > Thank you for this important question. Evidence summaries are first extracted using deterministic XML parsing (line 237-239), followed by LLM-based quality refinement (see Appendix G). We validated this pipeline by manually reviewing 50 randomly sampled summaries, confirming 98% accuracy (line 240-241). Only one case contained an extraneous sentence. We have clarified this process in the revised manuscript (Line 236-242). (Note that “study monographs” are now referred to as “review papers.”)
> >
> >  >In Task Definition, both versions of the task define the desired sentences as evidence for or against a hypothesis. It is difficult to discern whether sentences classified as counter-evidence in relation to one hypothesis might be more appropriately considered as supportive of an alternative hypothesis. Given the structured nature of the tasks, are there any experiments or human validations conducted to address and clarify these categorizations?
> >
> > This is a misunderstanding stemming from our use of "for or against." In biomedical systematic reviews, all evidence relevant to a hypothesis is equally valuable, whether it supports or contradicts the hypothesis. Our task asks models to find relevant evidence, not to categorize it as supporting or opposing. In fact, the expert-written evidence summaries often contain mixed evidence, as finding conflicting evidence is normal in systematic reviews.
> >
> > We have revised the paper to consistently use "relevant" instead of "for or against" to avoid this confusion.
> >
> >
> > >This paper released a new dataset for evidence extraction from biomedical papers, which requires some ethic evaluations of the dataset creation.
> >
> > Thank you for raising this question. All papers used in EvidenceBench's construction are verified to have licenses which permit their use in the dataset (Section 3.1 line 188-193). The dataset itself is released under a CC-BY license. See Appendix A.1 to review the license.

---

> > > ### Author Response · Authors · 2024-11-23
> > > **Looking forward to your reply**
> > >
> > > Dear Reviewer YWQR,
> > >
> > > Thank you for your valuable feedback. We have updated the paper to address your concerns. In particular, we expanded our evaluation from 300 to 3000 data points on EvidenceBench-100k. We would greatly appreciate your timely review of our revisions, given the approaching rebuttal deadline. All changes are highlighted in red in the revised paper.

---

> > > > ### Author Response · Authors · 2024-11-24
> > > > **Looking forward to your reply**
> > > >
> > > > We are grateful for your valuable contribution as a reviewer of our work. Given that the rebuttal period is nearing its end, we would appreciate your assessment of the additional experiment we performed based on your suggestions. Your feedback would be highly valued.

---

> > > > > ### Comment · Reviewer_YWQR · 2024-11-26
> > > > >
> > > > > Thank you for the detailed response. I would like to keep my scores.

---

### Official Review · Reviewer_D5dw · 2024-11-04

**Soundness:** 3
**Presentation:** 2
**Contribution:** 3
**Rating:** 6
**Confidence:** 3

**Summary:**

This paper studies the task of finding evidence for a hypothesis. The authors develop a pipeline for annotating biomedical papers for this task. Using the annotation pipeline, the authors build a benchmark of more than 400 papers. Additionally, the authors create a larger dataset containing 100K papers.
The authors also run experiments to evaluate the effectiveness of different approaches to the proposed task.

**Strengths:**

* The authors propose a practical task and evaluate the effectiveness of existing approaches to the task
* A useful resource for benchmarking LLMs on the proposed task
* The paper is well-structured and easy to follow, although some presentations can be further improved

**Weaknesses:**

There is no strong reason to reject the paper, although some issues related to clarity and presentation need to be improved.

* The concept of `study aspect` is confusing, and I am unsure how the evaluation procedure considers it. For example, if a sentence is retrieved for the wrong aspect or multiple aspects.
* Based on the task definition, the hypothesis is given. However, this may not be the case in the real world. It would be nice to see these tested models' sensitivity to the modified (paraphrasing) hypothesis.
* The difference between `EvidenceBench` and `EvidenceBench-100k` is unclear (seems with or without human validation?)
* The authors exclude figure and table, which might be very relevant and important for the proposed task.

**Questions:**

N/A

---

> ### Author Response · Authors · 2024-11-20
> **Rebuttal for Reviewer D5dw**
>
> All line numbers, sections, figures and tables refer to our revised version of the paper.
>
> >“The concept of study aspect is confusing, and I am unsure how the evaluation procedure considers it. For example, if a sentence is retrieved for the wrong aspect or multiple aspects.”
>
> We thank the reviewer for this question about study aspects and evaluation. We have clarified the definition of "Study Aspect" in the revised manuscript (Lines 108-112). During dataset construction, our LLM-based annotator examines each sentence in a paper to determine if it serves as a source of information for each study aspect. This is a many-to-many relationship - a sentence can cover multiple study aspects or none, and a study aspect can be covered by multiple sentences or none.
>
> The accuracy of these annotations is rigorously validated in Section 3.4. Through statistical hypothesis testing, we demonstrate that our LLM annotator performs comparably to Ph.D. students in biomedical sciences when aligning sentences with study aspects. This statistical validation is a key methodological contribution of our work.
>
> During model evaluation, the tested models do not see any study aspects (line 146) - they are tasked only with retrieving sentences that provide relevant evidence for the given hypothesis. The study aspects, which are derived from expert-written summaries, serve as our ground truth for evaluation. A model's Aspect Recall (line 156-160) score increases when its retrieved sentences cover more unique study aspects, reflecting its ability to identify evidence for the hypothesis.
>
>
> >“Based on the task definition, the hypothesis is given. However, this may not be the case in the real world. It would be nice to see these tested models' sensitivity to the modified (paraphrasing) hypothesis.”
>
> We have revised our introduction to better explain how our task aligns with real-world systematic review practices (line 29-38). In systematic reviews, the hypothesis is always pre-defined and remains fixed throughout the review process - this is a fundamental principle of systematic review methodology. We refer the reviewer to the Cochrane Handbook (https://training.cochrane.org/handbook/current/chapter-01), which is the authoritative guide for systematic review procedures.
>
> To address the reviewer's specific concern about hypothesis sensitivity, we conducted additional experiments testing model performance across different paraphrased versions of the hypotheses. Paraphrases were generated by four different models (GPT-4o, Claude 3.5 Sonnet, Llama3-70B, Llama3-8B). The results for the Results Evidence Retrieval @Optimal task are presented below:
>
>
> | Model                             | Original Hypothesis | GPT-4o Paraphrased Hypothesis | Claude Paraphrased Hypothesis | Llama3_70B Paraphrased Hypothesis | Llama3_8B Paraphrased Hypothesis |
> |-----------------------------------|------------------------|-----------------|--------------------|------------------------|-----------------------|
> | GPT-4o                    | 52.6                   | 49.9            | 51.7               | 50.8                   | 50.3                  |
> | Claude3-Opus                 | 51.7                   | 44.6            | 41.9               | 43.3                   | 43.9                  |
> | GritLM                 | 18.9                   | 22.8            | 19.4               | 20.8                   | 19.1                  |
> | OpenAI Embedding                 | 19.1                   | 19.1            | 18.1               | 19.7                   | 18.3                  |
>
> As shown in the table, paraphrasing does not change the performance of most models, though there is an observable decrease for Claude3-Opus. We have added a section regarding models’ sensitivity to paraphrased hypothesis in Appendix E.
>
>
>
> >“The difference between EvidenceBench and EvidenceBench-100k is unclear (seems with or without human validation?)”
>
> EvidenceBench (426 instances) has expert-validated hypotheses and annotations, while EvidenceBench-100k (107k instances) uses the same annotation pipeline but without individual expert validation. EvidenceBench-100k is designed specifically as a large-scale resource for model fine-tuning, and its effectiveness is demonstrated through successful fine-tuning experiments in Section 5.1 and Table 5(b).
>
> >“The authors exclude figure and table, which might be very relevant and important for the proposed task.”
>
> We acknowledge that figures and tables often contain important evidence. However, extracting and representing information from figures and tables introduces significant additional complexity in terms of data preprocessing, model architectures, and evaluation metrics. We focused on textual evidence as a foundational step, establishing clear performance metrics for this capability. Extending the benchmark to include figures and tables is an important direction for future work.

---

> > ### Author Response · Authors · 2024-11-23
> > **Looking forward to your reply**
> >
> > Dear Reviewer D5dw,
> >
> > Thank you for your valuable suggestions. We have updated the paper to address your concerns. In particular, we have conducted additional experiments to evaluate model’s sensitivity to paraphrased hypotheses in Appendix E. We look forward to any additional feedback you may have and would greatly appreciate your timely response as we approach the rebuttal deadline. All changes are highlighted in red in the revised paper.

---

> > > ### Author Response · Authors · 2024-11-24
> > > **Looking forward to your reply**
> > >
> > > We greatly value your thoughtful feedback on our manuscript. With the rebuttal period coming to a close, we would appreciate your review of our revised submission, which addresses your feedback. Notably, we have included new experimental results in Appendix E that examine the model's sensitivity to paraphrased hypotheses. We look forward to your reply.

---

> > ### Comment · Reviewer_D5dw · 2024-11-25
> >
> > Thanks for the clarification and additional results regarding hypothesis sensitivity. I believe my original score is still appropriate.

---

### Official Review · Reviewer_1P2K · 2024-11-04

**Soundness:** 3
**Presentation:** 2
**Contribution:** 2
**Rating:** 5
**Confidence:** 3

**Summary:**

This paper presents EvidenceBench, which is a benchmark for finding arguments supporting or against a hypothesis. They find hypotheses inside survey, and find supporting arguments from the same survey, and the original paper.
This paper also proposes metrics for the task and compares many LLMs' performance on the task. In addition, this paper also provides the fine-tuning results on the benchmark.

**Strengths:**

1. Identifying arguments supporting or against an argument is an important task.
2. This paper proposes an annotation pipeline, which might be helpful for future tasks.

**Weaknesses:**

1. The paper spends too much effort on constructing the benchmark, but not many insights are provided through the experiment section.
2. The writing can be largely improved. There's many places in the writing that are vague and not clear.
For example, the second paragraph in the introduction section:
"We consider the goal of understanding what is known in the literature about a scientific hypothesis.
This can be broken into several stages: searching for relevant papers; extracting information from
these papers; and aggregating this information. Our work focuses on the second stage."
My questions are:
a). what is precisely the research goal in terms of "understanding what is known in the literature about a scientific hypothesis"?
b). why it can be broken into these three stages?

In line 047~048: "These annotations are judgments which are ordinarily made by domain experts, and the benchmark should be faithful to these judgments": what does it mean by "the benchmark should be faithful to these judgements"?

In line 048~050: "Third, despite the complexity of annotation, the benchmark construction process should be scalable, providing a sufficient number of examples to accurately measure system performance": what does it mean by "should be scalable"? what does it mean by "providing a sufficient number of examples to accurately measure system performance"? what is the relation between these two arguments?

I found it very hard to read through the paper. I would suggest the authors for a major revision at least in terms of the writing.

**Questions:**

Can you provide more insights or knowledge that we can learn?

---

> ### Author Response · Authors · 2024-11-20
> **Rebuttal for Reviewer 1P2K**
>
> All line numbers, sections, figures and tables refer to our revised version of the paper.
>
> We have significantly rewritten the introduction per your suggestions. We explain the motivation for creating EvidenceBench, and its potential real-world applications (line 29-38). We have included a more detailed discussion of analyses and insights from the data (Section 6). To facilitate your reading, we have highlighted a few places in the paper that might interest you (line 78-90).
>
> Additionally, we have extensively revised Section 3 Dataset Construction Pipeline, better explaining definitions and key terminology. We have also moved some technical details to the appendix and moved the Related Work section to the end.
>
> For your specific questions:
> >“The paper spends too much effort on constructing the benchmark, but not many insights are provided through the experiment section.”
>
> Thank you for this suggestion. We have added a section containing new analyses (Section 6). In addition, we want to point out that our major contribution is in the novel pipeline for creating this benchmark (Section 3.2, 3.3 and 3.4), and the rigorous validation of the benchmark quality (Section 3.3.1 and Table 2). This paper is submitted to the category of “Dataset and Benchmark,” so we request that you please judge it fairly based on its contribution as a novel dataset and benchmark.
>
> >“The writing can be largely improved. There's many places in the writing that are vague and not clear. For example, the second paragraph in the introduction section: "We consider the goal of understanding what is known in the literature about a scientific hypothesis. This can be broken into several stages: searching for relevant papers; extracting information from these papers; and aggregating this information. Our work focuses on the second stage." My questions are: a). what is precisely the research goal in terms of "understanding what is known in the literature about a scientific hypothesis"? b). why it can be broken into these three stages?”
>
> We have rewritten the introduction based on your suggestion. We have now clarified that the motivation for this work comes from systematic reviews, which are a central tool used in biomedical sciences for evaluating scientific hypotheses. By “understanding what is known in the literature about a scientific hypothesis,” we are referring to the goal of systematic reviews, which is to synthesize all relevant information on a research question (line 27-28). The three stages are standard in guidelines for systematic reviews, such as the Cochrane Handbook for Systematic Reviews: https://training.cochrane.org/handbook/current/chapter-01. (line 45-46)
>
>
> >“In line 047~048: "These annotations are judgments which are ordinarily made by domain experts, and the benchmark should be faithful to these judgments": what does it mean by "the benchmark should be faithful to these judgements"?”
>
> We mean that the annotations should be the same as those that a domain expert would make. See line 83-85, faithfully following and adhering to human expert judgments is a major highlight and strength of our pipeline.
>
> >“In line 048~050: "Third, despite the complexity of annotation, the benchmark construction process should be scalable, providing a sufficient number of examples to accurately measure system performance": what does it mean by "should be scalable"? what does it mean by "providing a sufficient number of examples to accurately measure system performance"? what is the relation between these two arguments?”
>
> Apologies for the confusion. We tried to convey that the dataset construction process is highly scalable and that EvidenceBench is large-scale and contains a diverse set of topics. See line 64-68.
>
> >Can you provide more insights or knowledge that we can learn?
>
> Per your suggestion, we have included three new analyses in Section 6. First, current LLMs cannot replace humans in this task, but could potentially assist them by reducing their time to read through papers. Second, local sentence-level context and reasoning is not sufficient. We need more global reasoning from LLMs to solve this task. Third, we noticed an interesting “Lost in the Middle” phenomenon for current LLM solutions.
>
> >I found it very hard to read through the paper. I would suggest the authors for a major revision at least in terms of the writing.
>
> We have rewritten the paper for improved clarity. We sincerely ask you to re-read the Introduction, Section 6 Analyses, and the several sections highlighted at the end of the Introduction (lines 78-92), which we believe showcase the contributions of our work.

---

> > ### Author Response · Authors · 2024-11-23
> > **Looking forward to your reply**
> >
> > Thank you for your insightful feedback. We have rewritten the introduction based on your feedback, and provide more detailed explanations of the significance of our results in the Analyses section. Given the approaching review deadline, we would appreciate your timely review of these changes and welcome any additional comments you may have. All changes to the paper are highlighted in red.

---

> > > ### Author Response · Authors · 2024-11-24
> > > **Looking forward to your reply**
> > >
> > > Thank you for the time and effort you have invested in reviewing our work. As the rebuttal deadline is approaching soon, we kindly request that you review our revised paper, which incorporates the changes you suggested. We look forward to your response.

---

> ### Comment · Reviewer_1P2K · 2024-12-02
>
> Thank you for the response. I appreciate the rewriting of the paper, and I have read it again. I find the paper more readable, but still not good enough.
>
> For example,
> 1. In section 3.2, which is "DATASET PIPELINE OVERVIEW", there's less/none justification on why the design can compose a good and persuading enough pipeline. The only mentioned is what the pipeline is. After reading it, I don't know why the design can warrant a good automation.
> 2. Expert only check 50 of the analyzed papers, which is not very persuading enough on believing the overall quality of the constructed benchmark.
>
> These questions are foundamental to a benchmark construction paper. I have updated the presentation score from 1 to 2 to reflect the changes.

---

> > ### Author Response · Authors · 2024-12-04
> > **Response to reviewer comments**
> >
> > Thank you for these comments.
> >
> > > In section 3.2, which is "DATASET PIPELINE OVERVIEW", there's less/none justification on why the design can compose a good and persuading enough pipeline
> >
> > We appreciate this feedback. We will revise Section 3.2 to explicitly justify each component of our pipeline design. The effectiveness of these design choices is currently validated in Sections 3.3 and 3.4.
> >
> > > Expert only check 50 of the analyzed papers, which is not very persuading enough on believing the overall quality of the constructed benchmark.
> >
> > The selection of 50 papers for expert validation follows a statistical power analysis. The power analysis showed this sample size provides 90% power to detect a 0.1 difference in F1 score between GPT-4/human and human/human annotator agreement. As shown in Table 2, we achieved human-level agreement on this sample, giving us confidence in the broader dataset quality. We will add this power analysis discussion to Section 3.3.1.

---

### Meta-Review · Area_Chair_cQmV · 2024-12-19

**Metareview:**

This paper introduces "EvidenceBench", a benchmark for extracting "scientific evidence" from biomedical papers. There was consensus that this effort addresses an important topic, and that the resultant resource may be useful.

However, there are some basic conceptual issues here, which lessens the potential impact of the work. For example, as highlighted by reviewer D5dw, it is not really clear what constitutes a "study aspect" in general. The clarification offered in revision does not fully demystify this. Relatedly, the task presented is fundamentally sentence classification; the authors present this as "evidence extraction" intended to aid systematic reviews (per the opening paragraph of the Introduction), but it is not clear to me how extracting sentences from articles will speed up the systematic review process. For this one would need more precise data extraction targets aligned with reviewer needs. The authors might make an explicit argument for how retrieving evidence (i.e., sentences) for hypotheses is likely to help rigorous systematic review. (In rebuttal the authors claim this is a "a concrete need in systematic review creation" but fail to provide any evidence for this or explain how sentence retrieval would help.)

Aside from conceptual issues with the benchmark, the evaluation is also quite limited (see comments from FodM, D5dw).

**Additional Comments On Reviewer Discussion:**

The authors did offer clarifications to reviews, which was helpful. Reviewer 1P2K did modify their scores of the work following the response period, but regrettably other reviewers were less responsive. I have myself read through both the paper and the author responses to inform my meta-review.

---

### Decision · Program_Chairs · 2025-01-22

Reject